

# New probabilistic methods for quantitative climate reconstructions applied to palynological data from Lake Kinneret

Timon Netzel[1], Andrea Miebach[2], Thomas Litt[2], and Andreas Hense[1]

[1]Institute for Geoscience, Meteorology, University of Bonn, Auf dem Hügel 20, 53121 Bonn, Germany
[2]Institute for Geoscience, Paleontology, University of Bonn, Nussallee 8, 53115 Bonn, Germany

**Correspondence:** Timon Netzel (timon.netzel@uni-bonn.de)

**Abstract.** Quantitative local paleoclimate reconstructions are an important tool for gaining insights into the climate history of the Earth. The complex age–sediment–depth and proxy–climate relationships must be described in an appropriate way. Bayesian hierarchical models are a promising method for describing such structures.

In this study, we present a new age–depth transformation in a Bayesian formulation by determining the uncertainty infor-
mation of depths in lake sediments at a given age. This enables data-driven smoothing of past periods, which allows for better interpretation.

Furthermore, we introduce a systematic way to establish transfer functions that map climate variables to biome distributions. This includes consideration of various machine learning algorithms for solving the classification problem of biome presence and absence, taking into account uncertainties in the proxy–climate relationship. For the models and biome distributions used,
a simple feedforward neural network wins.

Based on this, we formulate a new Bayesian hierarchical model that generates local paleoclimate reconstructions. This is applied to plant-based proxy data from the lake sediment of Lake Kinneret. Here, a priori information on the recent climate in this region and data on arboreal pollen from this lake are used as boundary conditions. To solve this model, we use Markov chain Monte Carlo sampling methods. During the inference process, our new method generates taxa weights and biome climate
ranges. The former shows that less weight needs to be given to *Olea europaea* to ensure the influence of the other taxa. In contrast, the highest weights are found in *Quercus calliprinos* and Amaranthaceae, resulting in appropriate flexibility under the given boundary conditions. In terms of climate ranges, the posterior probability of the Mediterranean biome reveals the greatest change, with an average boreal winter (December–February) temperature of $10\,^{\circ}\mathrm{C}$ and an annual precipitation of $700\,\mathrm{mm}$ for Lake Kinneret during the Holocene. The paleoclimate reconstruction for this period shows comparatively low precipitation of
about $400\,\mathrm{mm}$ during 9–7 and 4–2 cal ka BP. The respective temperature fluctuate much less and stays around $10\,^{\circ}\mathrm{C}$.

## 1 Introduction

Local paleoclimate reconstructions reveal information of the climatic history of relatively small regions. In the last few decades, a lot of reconstructions were published, which showed the advantages and disadvantages of the respective methodologies. One promising way is the idea of the indicator species approach, which is the basic of the model used in this study. Here, plant



distribution maps are linked to recent climate data to define a climate range where the corresponding taxon occurs. Finally, when considering multiple taxa, these climatic ranges can be combined to determine the mutual climatic range (MCR).

We follow the idea of Kühl et al. (2002), who developed a probabilistic interpretation of MCR. This addresses the problem of overfitting by calculating uncertainty ranges for each taxon. These were initially based on two- or three-dimensional Gaussian probability density functions (PDFs), which is why this is called the PDF method. This basic concept was extended and applied

for both local and spatial climate reconstructions. For example, Kühl and Litt (2003) calculated January and July temperatures for three sites in Central Europe during the last Interglacial period. Subsequently, spatial reconstructions of Europe were performed in Gebhardt et al. (2008) for the Eemian, in Simonis et al. (2012) for the late Glacial and Holocene, and in Weitzel et al. (2019) for the mid-Holocene (MH). Over time, more complex machine learning methods such as the Generalized Linear Model and Quadratic Discriminant Analysis (QDA) are used to determine the transfer functions (e.g. Litt et al., 2012; Weitzel

et al., 2019). Schölzel (2006) describes the PDF method in the context of a Bayesian hierarchical model (BHM) and calls it Bayesian Indicator Taxa Model (BITM). This has the advantage that additional prior information can regulate the transfer functions and thus the entire climate reconstruction. Among others, the BITM was applied in Neumann et al. (2007) for Birkat Ram in Israel and in Thoma (2017) for Lake Prespa in Greece.

The basic of the climate reconstruction used in this work is first presented by Schölzel (2006). This is another BHM, the so-

called Bayesian Biome Model (BBM). In this process, certain plant taxa are assigned to different biomes. These are groups of taxa that have similar vegetation zones under comparable climatic conditions (Prentice et al., 1992). One advantage of the BBM is that no recent distribution maps for every plant occurring in the core is needed, but only for the biomes used. Applications of this model can be found in Schölzel (2006), Litt et al. (2012), and Stolzenberger (2017) for the Dead Sea. Thoma (2017) applied the BBM to Lake Kinneret (LK), also known as the Sea of Galilee, with the result that the two biomes used showed

too little variability and suggested an expansion to at least three biomes. The BBM also allows reconstructions based on prior climate data. These come, for example, from other studies that suggest possible climate ranges for the reconstruction site and period (Schölzel, 2006). Once set, they cannot be adjusted during the reconstruction process.

Although the above-mentioned methods for reconstructing the local climate are already quite well elaborated, they still have some disadvantages:

1. full age uncertainties are not taken into account

2. manual selection of which information about the plants are to be included in the model

3. to obtain different reconstructions, for example, certain taxa have to be removed or added manually

4. in order to diagnose the occurrence of certain taxa, a priori thresholds, e.g. on the number of pollen counts, have to be set

5. no consideration of human impact on the vegetation surrounding the study site

6. a flexible spatio-temporal adjustment of the transfer functions and thus the reconstruction is not possible



7. the need for a more systematic method to determine the most appropriate model to describe the relationship between plant and climate data.

The last point is due to the model assumption that the relationship between recent climate and plant distribution has not changed during the reconstruction period.

The aim of this study is to develop an algorithm that provides potential and comprehensible solutions to the problems summarized above. The traceability of the proposed method for calculating quantitative paleoclimate reconstructions results from increased automation and statistical modelling rather than from additional assumptions. This provide new insights into the importance of the proxies studied and thus extend the knowledge from previous studies. We want to apply the new model to botanical proxy data (pollen and macrofossils) from LK. Based on multiple proxy information, there are a variety of environmental and therefore qualitative climate reconstructions in the vicinity of LK (Schiebel and Litt, 2018; Miebach et al., 2022; Orland et al., 2009). We will use these as a comparison for our quantitative statements to show similarities and differences and to check whether our new approach fits into the existing knowledge and thus provides realistic results.

The structure of this work is as follows. In Sect. 2 we give an overview of the study area. The following Sect. 3 first deals with the botanical data used. The individual modules of the proposed Bayesian framework are then described in detail. Sect. 4 presents the results of our new reconstruction method. These are then discussed, summarized, and possible extensions suggested in Sects. 5 and 6.

## 2  Study area

The location of Lake Kinneret is marked with a black dot in Fig. 1 (a). LK is a warm, monomictic and meso-eutrophic inland lake being part of the Jordan river catchment and its lake level varies between 209 and 215 m below mean sea level. It has a maximum water depth of ca. 42 m and a surface area of ca. 169 km$^2$ ($21 \times 12$ km at the maximum). The watershed area comprises 2730 km$^2$ (Berman et al., 2014).

The Sea of Galilee occupies the LK basin along the active Dead Sea fault. It developed by several tectonic processes (Ben-Avraham et al., 2014). The regional morphology is characterized by Cretaceous to Eocene carbonate rocks with extensive karst and Neogene and Pleistocene basalts (Sneh et al., 1998). Soils such as terra rossa and rendzina form the surface cover of the Galilee Mountains (Dan et al., 1972). Alluvial and lacustrine sediments of Pleistocene to Holocene ages fill the Jordan Valley north and south of the Sea of Galilee (Sneh et al., 1998).

Fig. 1 (a) and (b) show the spatial distribution of the mean December–February temperature ($T_{DJF}$) and annual precipitation ($P_{ANN}$) that we will examine in more detail in this study. In particular, this means that they will be reconstructed for the LK. The Mediterranean climate with hot, dry summers and mild, wet boreal winters is typical of northern Israel, as shown by the Koeppen–Geiger classification Csa in the climate diagram of LK in Fig. 1 (c). The basin of the lake is characterized by 400 mm mean annual precipitation and 21 °C mean annual temperature. The surrounding mountains, however, experience $P_{ANN}$ rates of up to $> 900$ mm and annual temperatures of less than 15 °C. The climate diagram reflects these relatively large variations, which result from the $0.5° \times 0.5°$ horizontal resolution of the Climate Research Unit (CRU) data (Harris et al., 2020). 90 %



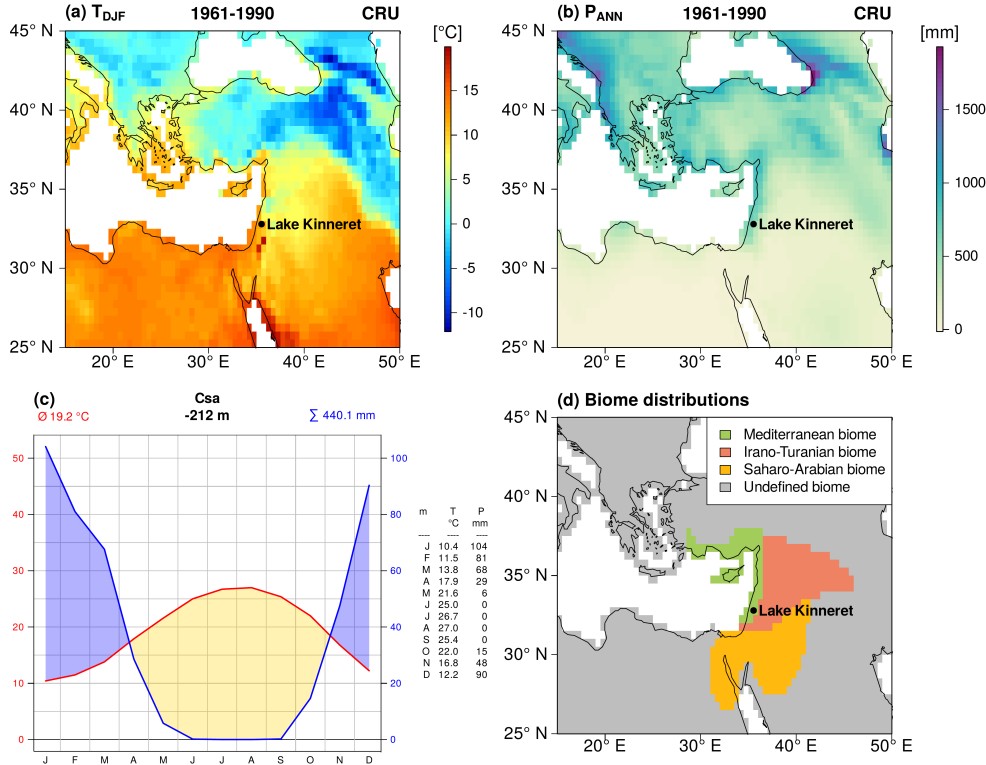

**Figure 1.** The mean December–February temperature $T_{DJF}$ in (a) and annual precipitation $P_{ANN}$ in (b) based on the current version CRU 4.07 data and the period 1961–1990. The black dot marks the location of LK. A climate diagram of the grid point closest to LK is shown in (c), based on these CRU data. Panel (d) depicts the biome definitions used in this work.

of the precipitation in the north of Israel comes from so-called Cyprus lows, that form over the eastern Mediterranean. These mainly occur from October to May, with the heaviest rainfall between December and March (Ziv et al., 2014).

    Furthermore, Fig. 1 (d) shows the biome distributions considered in this work. The colored areas distinguish the following biomes: The Mediterranean, the Irano-Turanian, the Saharo-Arabian, and the unspecified biome. We can see that the majority of the lake's watershed can be ascribed to the Mediterranean biome, while the southern lakeshore borders the Irano-Turanian

biome (Zohary, 1962). The Mediterranean biome is distributed in areas exceeding $300\,\mathrm{mm}$ of $P_{ANN}$ (cf. Fig. 1 (b)). The climax vegetation is dominated by trees and shrubs. Typical plants are *Quercus ithaburensis, Q. boisseri, Q. calliprinos, Olea europaea, Pistacia lentiscus, Arbutus andrachne, Ceratonia siliqua, Pinus halepensis*, and *Sarcopoterium spinosum* (Danin, 1988; Zohary, 1982). The Irano-Turanian steppe grows in areas below $300\,\mathrm{mm}$ of $P_{ANN}$ (cf. Fig. 1 (b)). The biome is rich in semi-shrubs, annual herbs, and geophytes. Common taxa are *Artemisia herba-alba, Thymelaea hirsute*, and various Poaceae

and Amaranthaceae (including Chenopodiaceae) (Danin, 1992; Zohary, 1982)



## 3 Material and methods

### 3.1 Material

The material used in this study originates from lacustrine sediment cores from the central Sea of Galilee. They were recovered in March 2010 during a drilling campaign within the Collaborative Research Center 806 "Our Way to Europe" funded by the

German Research Foundation (DFG). Two parallel cores (Ki10 I with $13.3\,\mathrm{m}$ core recovery and Ki10 II with $17.8\,\mathrm{m}$ core recovery) were obtained at a water depth of $38.8\,\mathrm{m}$. Both cores were combined to a $17.8\,\mathrm{m}$ composite profile. Besides a $25\,\mathrm{cm}$ varved sequence at the top, the sediment comprises homogeneous grayish-brownish silts and clays (Schiebel and Litt, 2018).

### 3.2 Palynology

Additional samples were added to the palynological datasets by Schiebel and Litt (2018) and Langgut et al. (2013) to increase

the temporal resolution. The resulting dataset consists of 160 samples with a mean resolution of $11\,\mathrm{cm}$. We followed a standard preparation technique by Faegri and Iversen (1989) to extract pollen from the lake sediment (see Schiebel and Litt (2018) for more details). At least 500 terrestrial pollen grains were identified under a light microscope at $400\times$ magnification with the help of the pollen reference collection from the Institute of Geosciences, University of Bonn, and pollen atlases (Reille, 1995, 1998, 1999; Beug, 2004). Pollen percentages are based on the terrestrial pollen sum excluding indeterminable pollen

grains and obligate aquatic plants to exclude local taxa growing in the lake (Moore et al., 1991). Pollen zonation was adapted from Schiebel and Litt (2018).

### 3.3 Relationship between age and depth

#### 3.3.1 Age–depth model

We start with the Bayesian based age–depth model from Miebach et al. (2022) to describe the relationship between age and

depth. It provides a probabilistic model of the sediment accumulation rate of the core necessary to reach the $^{14}\mathrm{C}$ ages at the available depths within the dating uncertainties. We use the Bacon model implemented in R (R Core Team, 2018; Blaauw et al., 2020). This is explained in detail in Blaauw and Christen (2011) and is only briefly described in the following.

Bacon uses a self-adjusting Markov chain Monte Carlo (MCMC) simulation to calculate the conditional probability distribution $\mathbb{P}(\boldsymbol{\phi}, \boldsymbol{r}, \boldsymbol{m}|\boldsymbol{x})$, where $\boldsymbol{\phi}$ contains the model parameter, $\boldsymbol{r}$ the accumulation rate, $\boldsymbol{m}$ the memory, and $\boldsymbol{x}$ the measurements

such as $^{14}\mathrm{C}$ data. More precisely, this distribution describes the posterior (conditional) probabilities of $\boldsymbol{\phi}, \boldsymbol{r}$, and $\boldsymbol{m}$ given the age data $\boldsymbol{x}$ at depth $\boldsymbol{D}$. However, as we will see in the next section, we are actually interested in the conditional probability of depth $\boldsymbol{D}$ at a fixed age $\boldsymbol{A}$, which can be derived using the accumulation rates $\boldsymbol{r}$ and applying Bayes' theorem.

#### 3.3.2 Age–depth transformation

Now we will consider how to use the probabilistic information from models like Bacon to calculate a transformation from

depth to age. For this purpose, it is useful to look at Table 1, which describes the variables used in this work. Using Bayesian



**Table 1.** List of variables.

| Variable shortcut | Description |
|---|---|
| $\mathbb{P}$ | Probability distribution |
| $\boldsymbol{C}$ | Climate: contains modern $\boldsymbol{C_m}$ and past $\boldsymbol{C_p}$ climate information |
| $\boldsymbol{P}$ | Proxy: contains modern $\boldsymbol{P_m}$ and past $\boldsymbol{P_p}$ proxy information |
| $\boldsymbol{P_s}$ | Selected plant proxy information |
| $\boldsymbol{PP}$ | Proxy pool: explained variances between additional proxies and $\boldsymbol{C}$ |
| $\boldsymbol{B}$ | Biome information |
| $\boldsymbol{A}$ | Age of, e.g. lake sediments |
| $\boldsymbol{D}$ | Depth of, e.g. lake sediments |
| $\boldsymbol{\Theta}$ | Contains the following parameters: |
| $\psi$ | Link between $\boldsymbol{C}$ and $\boldsymbol{B}$ |
| $\boldsymbol{\omega}$ | Contains all information about the taxa weights |

hierarchical modelling techniques, we can determine the joint probability density function (or probability mass function in the case of discrete random variables) of the target variables $\boldsymbol{Y}$, the age $\boldsymbol{A}$, the proxy data $\boldsymbol{P}$, and the required additional parameters $\boldsymbol{\Theta}$. These are of course all dependent on depth, but $\boldsymbol{D}$ is only an auxiliary variable due to the coring procedure. Therefore, the full joint probability density/mass function that includes $\boldsymbol{D}$ can be marginalized (integrated) with respect to

$\boldsymbol{D}$. In a second step, we apply the relationship between full, joined and the necessary conditional ones. This establishes the following equation:

$$\mathbb{P}(\boldsymbol{Y}, \boldsymbol{A}, \boldsymbol{P}, \boldsymbol{\Theta}) = \int_{\mathcal{D}} \mathbb{P}(\boldsymbol{Y}, \boldsymbol{A}, \boldsymbol{P}, \boldsymbol{D}, \boldsymbol{\Theta}) \, d\boldsymbol{D}$$

$$= \int_{\mathcal{D}} \mathbb{P}(\boldsymbol{Y} \mid \boldsymbol{A}, \boldsymbol{P}, \boldsymbol{D}, \boldsymbol{\Theta}) \cdot \mathbb{P}(\boldsymbol{D} \mid \boldsymbol{A}, \boldsymbol{P}, \boldsymbol{\Theta}) \cdot \mathbb{P}(\boldsymbol{A}, \boldsymbol{P}, \boldsymbol{\Theta}) \, d\boldsymbol{D}. \tag{1}$$

$\boldsymbol{Y}$ contains the variables we are interested in, e.g. $\boldsymbol{C}$. Now suppose that $\boldsymbol{D}$ is conditionally independent of $\boldsymbol{P}$ and $\boldsymbol{\Theta}$ and thus

fully dependent on $\boldsymbol{A}$. This is exactly the information we get from the age–depth relationship. Furthermore, the variables $\boldsymbol{Y}$ should not depend on age if $\boldsymbol{D}$ is given. This assumption follows from the fact that initially any information drawn from the sediment core are with respect to depth. Using this, we can transform Eq. 1 as follows:

$$\mathbb{P}(\boldsymbol{Y} \mid \boldsymbol{A}, \boldsymbol{P}, \boldsymbol{\Theta}) = \int_{\mathcal{D}} \mathbb{P}(\boldsymbol{Y} \mid \boldsymbol{P}, \boldsymbol{D}, \boldsymbol{\Theta}) \cdot \mathbb{P}(\boldsymbol{D} \mid \boldsymbol{A}) \, d\boldsymbol{D}. \tag{2}$$

As we can see we need a tool for the calculation of $\mathbb{P}(\boldsymbol{D}|\boldsymbol{A})$. This was developed for this work and can be found in the rbacon

package under the function *Bacon.d.Age*. Bacon calculates the slopes (accumulation rates) of a series of flexible linear age–depth functions. Their flexibility results from different $r$ in a priori defined regular sections along the depth axis. If a certain age





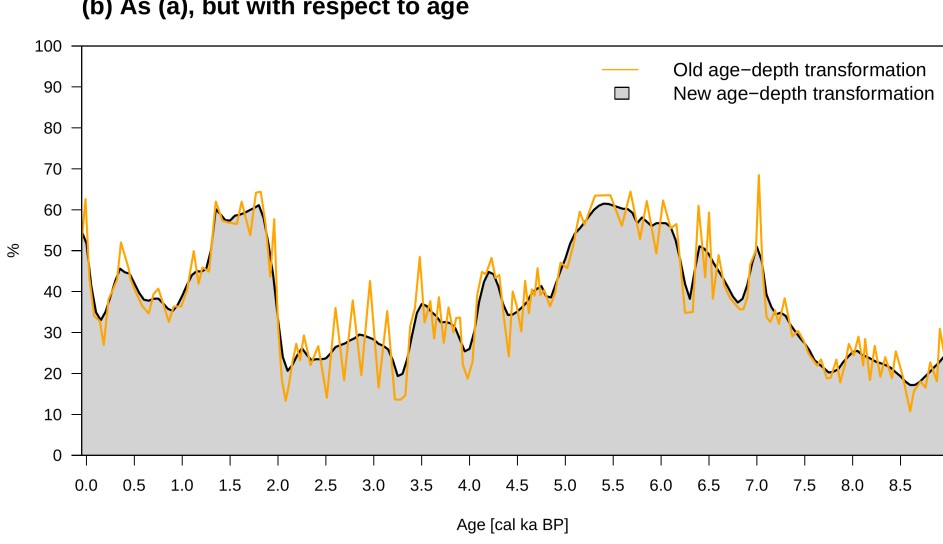

**Figure 2.** The gray areas show the percentages of aggregated arboreal pollen from LK. Panel (a) depicts these data in terms of depth and (b) in terms of age. On the one hand, in (b) we see the result when all probability information $\mathbb{P}(\boldsymbol{D} \mid \boldsymbol{A})$ are taken into account (gray area). The orange line, on the other hand, represents the output using the mean values of $\boldsymbol{A}$.

$a$ is specified, *Bacon.d.Age* searches for those sections where intersections between $a$ and the respective age–depth functions exist. In this way, we can calculate probability distributions of depths for each age within the reconstruction period.

$\mathbb{P}(\boldsymbol{D}|\boldsymbol{A})$ obtained in this way indicates which depth has a higher or lower (possibly approaching zero) probability of contributing to $\boldsymbol{Y}$ at a given age. Eq. 2 shows that the desired age-dependent target variables $\boldsymbol{Y}$ are calculated by a moving window






(convolution) stretching/compressing operation on the depth axis together with a smoothing of this axis at each sediment depth. The sliding windows are derived solely from the age–depth model data and do not necessarily follow a top-hat filter or any other smoothing function.

Fig. 2 illustrates the entire process using arboreal pollen (AP) from LK. In this case, the target variable is AP and is shown in panel (a) as a function of sediment depth. The mean age difference between the studied core intervals of $11\,\mathrm{cm}$ is 51 years. Thus, we define a regular temporal grid of 50 years via $\mathbb{P}(\boldsymbol{D}|\boldsymbol{A})$, resulting in a total of 181 age steps. Applying these to the data from panel (a) using Eq. 2, we get the result of the new age–depth transformation depicted in (b). In contrast, the orange line shows the result when the plant data in terms of depth are linked to the mean age data from the age–depth model. So far, this is a very common procedure (e.g. Litt et al., 2012; Schiebel and Litt, 2018; Torfstein et al., 2015; Neumann et al., 2007; Miebach et al., 2019; Seppä et al., 2005). The strongly fluctuating behaviour of this AP curve indicates an overfitting result, which makes interpretations difficult. With this new technique, we can circumvent such problems and have eliminated the first disadvantage mentioned in the introduction.

### 3.4 Reconstruction model

In this study, a reformulation of the BBM is used to calculate a quantitative climate reconstruction. A detailed derivation of the enhanced and extended BBM can be found in Appendix A. The final result can be written as:

$$\mathbb{P}(\boldsymbol{C} \mid \boldsymbol{P}, \boldsymbol{A}, \boldsymbol{\Theta}) \approx \int_{\mathcal{B}} \frac{\mathbb{P}(\boldsymbol{B} \mid \boldsymbol{C}, \boldsymbol{\psi}) \cdot \mathbb{P}(\boldsymbol{C} \mid \boldsymbol{\psi}) \cdot \mathbb{P}(\boldsymbol{\psi})}{\mathbb{P}(\boldsymbol{B}, \boldsymbol{\psi})} \cdot \int_{\mathcal{P}_s} \mathbb{P}(\boldsymbol{B} \mid \boldsymbol{P}, \boldsymbol{P_s}, \boldsymbol{\omega}) \cdot \mathbb{P}(\boldsymbol{P_s} \mid \boldsymbol{P}, \boldsymbol{A}, \boldsymbol{\omega}) \, d\boldsymbol{P_s} \, d\boldsymbol{B}. \tag{3}$$

This is the basic model calculated $\mathrm{N_{sample}}$ times with $i = 1, ..., \mathrm{N_{sample}}$ for different $\boldsymbol{\omega}_i$ and $\boldsymbol{C}_i$ by systematically sampling from the pools of plant information and transfer function distributions using MCMC techniques. In order to be able to describe this in more detail, certain framework conditions must be introduced. To this end, we will introduce reference curves based, for example, on AP data from lake sediments (see Fig. 2 (b)). If a reconstruction according to Eq. 3 is performed for certain $\boldsymbol{\omega}_i$ and $\boldsymbol{C}_i$, the resulting $\mathbb{P}(\boldsymbol{C}_i \mid \boldsymbol{P}, \boldsymbol{A}, \boldsymbol{\Theta}_i)$ can be compared with these reference curves. Here, the explained variances $\mathrm{R}^2$ are used as a similarity measure and stored in a variable we call proxy pool $\boldsymbol{PP}$. Based on this idea, an extended BHM can be constructed (the weighting term is omitted for convenience):

$$\mathbb{P}(\boldsymbol{C}, \boldsymbol{\Theta} \mid \boldsymbol{P}, \boldsymbol{A}, \boldsymbol{PP}) \propto \mathbb{P}(\boldsymbol{PP} \mid \boldsymbol{C}, \boldsymbol{P}, \boldsymbol{A}, \boldsymbol{\Theta}) \cdot \mathbb{P}(\boldsymbol{C} \mid \boldsymbol{P}, \boldsymbol{A}, \boldsymbol{\Theta}) \cdot \mathbb{P}(\boldsymbol{P} \mid \boldsymbol{A}, \boldsymbol{\Theta}) \cdot \mathbb{P}(\boldsymbol{A}, \boldsymbol{\Theta}). \tag{4}$$

At this point, one could add a variety of additional reference curves based on, for example, isotopes from lake sediments, marine sediments, ice core, solar radiation or $CO_2$ information. Some of them are used in Netzel (2023a), where they serve as guidelines for the calculation of several hundred paleoclimate reconstructions in the European region, using a technique similar to the one described here. However, only proxies derived from botanical information are considered in this work.

We know that some sections of the AP curves have fluctuations that are not due to climatological changes (e.g. Panagiotopoulos et al., 2013; Miebach et al., 2016; Neumann et al., 2007; Litt et al., 2012; Schiebel and Litt, 2018). In particular, the influence of humans on the vegetation around the lakes studied during the mid- to late Holocene complicates the interpretation of these curves. To account for such uncertainties, we specify a priori that the climate reconstructions should explain





about 50 % of the variance of the respective reference curves. This offers, on the one hand, the possibility to follow the trends of the references and supports the assumption that the biome information $B$ provide sufficient variability. On the other hand,

additional factors that might influence the references are allowed. Thus, we specify an independent prior proposal distribution for the explained variance with a mean of 0.5 and and a standard deviation of about 0.2. The latter gives our model the ability to capture a sufficiently large range of $R^2$ (Netzel, 2023a). Such a proposal can be described by a beta distribution with shape parameters three:

$$\mathbb{P}(\boldsymbol{PP} \mid \cdot) = \text{Beta}(\boldsymbol{PP} \mid 3, 3). \tag{5}$$

This is the first term on the right of Eq. 4, which we call the proxy pool module.

The second term can be analyzed as follows:

$$\mathbb{P}(\boldsymbol{C} \mid \boldsymbol{P}, \boldsymbol{A}, \boldsymbol{\Theta}) = \mathbb{P}(\boldsymbol{C_p} \mid \boldsymbol{C_m}, \boldsymbol{P}, \boldsymbol{A}, \boldsymbol{\Theta}) \cdot \mathbb{P}(\boldsymbol{C_m} \mid \boldsymbol{P}, \boldsymbol{A}, \boldsymbol{\Theta}). \tag{6}$$

$\mathbb{P}(\boldsymbol{C_p} \mid \cdot)$ gives us the ability to constrain the reconstructions based on additional climate information from the past. These can be, for example, other local reconstructions, paleoclimate simulations, or specific expert knowledge based on vegetation studies.

Consideration of such past climate data is shifted to future work, e.g. when high-resolution regional paleoclimate simulations become available. The second term on the right of Eq. 6 allows us to insert constraints on the reconstructed modern climate. We define the transition from modern times to the past at 0 cal a BP (calibrated years before the present), i.e. 1950 CE. This is because the temporal resolution of 50 years limits us, as we can only define the years 2000 CE or 1950 CE as the most recent period. For such a modern climate, we use the CRU data presented in Fig. 1 and create probability distributions as anchors for

the reconstructions. These independent proposal distributions are described by a normal distribution as an approximation for $T_{\text{DJF}}$ and a gamma distribution for $P_{\text{ANN}}$. All in all, we refer to the above as the prior climate module, which can be summarized as follows (with Unif being the uniform probability density):

$$\mathbb{P}(\boldsymbol{C_p} \mid \boldsymbol{C_m}, \boldsymbol{P}, \boldsymbol{A}, \boldsymbol{\Theta}) = \begin{cases} \mathbb{P}(\boldsymbol{C_p} \mid \cdot) & \text{if } \boldsymbol{C_p} \text{ is available,} \\ \text{Unif}(1, ..., N_{\text{age}}) & \text{otherwise,} \end{cases} \tag{7}$$

$$\mathbb{P}(\boldsymbol{C_m} \mid \boldsymbol{P}, \boldsymbol{A}, \boldsymbol{\Theta}) = \begin{cases} \Gamma(P_{\text{ANN},m}) \text{ and } \mathcal{N}(T_{\text{DJF},m}) & \text{if } \boldsymbol{A}(C_1) \leq 0 \text{ cal BP,} \\ \text{Unif}(1, ..., N_{\text{age}}) & \text{otherwise.} \end{cases} \tag{8}$$

This means that reconstructions can be carried out with fewer restrictions even without prior climate information. This is made possible by the use of uniform distributions that encompass the reconstruction period and thus all time slices $N_{\text{age}}$.

Finally, we consider the third term on the right of Eq. 4 in detail:

$$\mathbb{P}(\boldsymbol{P} \mid \boldsymbol{A}, \boldsymbol{\Theta}) = \mathbb{P}(\boldsymbol{P} \mid \boldsymbol{A}, \boldsymbol{\omega}, \boldsymbol{\psi}) \approx \mathbb{P}(\boldsymbol{P} \mid \boldsymbol{A}, \boldsymbol{\omega}) \approx \mathbb{P}(\boldsymbol{P} \mid \boldsymbol{\omega}). \tag{9}$$

First, we assume that the parameters $\boldsymbol{\omega}$ and $\boldsymbol{\psi}$ are a priori independent of each other. Then we state that $\boldsymbol{P}$ is independent of $\boldsymbol{\psi}$

if no $\boldsymbol{C}$ is given. Finally, the updated taxa weights $\mathbb{P}(\boldsymbol{P} \mid \boldsymbol{\omega})$ are determined under the assumption that they are conditionally





independent of A and thus hold for the entire reconstruction period. At this point, taxa weights could be split temporally based on additional prior information, so that they differ for specific time periods (e.g. glacials/interglacials). This approach is not explored further in this study and could be included in future work.

The last term of Eq. 4 is the joint distribution of $\boldsymbol{A}$ and $\boldsymbol{\Theta}$. We assume that all parameters $\boldsymbol{\Theta}$ are a priori independent of $\boldsymbol{A}$.
Thus, this distribution can be formulated as follows:

$$\mathbb{P}(\boldsymbol{A}, \boldsymbol{\Theta}) = \mathbb{P}(\boldsymbol{A}) \cdot \mathbb{P}(\boldsymbol{\psi}) \cdot \mathbb{P}(\boldsymbol{\omega}). \tag{10}$$

The second term contains the parameters of the transfer functions and $\boldsymbol{A}$ is assumed to be uniform distributed if no depth information are available. We see that already in the local reconstruction module in Eq. 3, where the relations between $\boldsymbol{A}$ and $\boldsymbol{D}$ are inserted into our reconstruction scheme. With all the reformulations and simplifications listed above, Eq. 4 can be
summarized as follows:

$$\mathbb{P}(\boldsymbol{C}, \boldsymbol{\Theta} \mid \boldsymbol{P}, \boldsymbol{A}, \boldsymbol{PP}) \propto \mathbb{P}(\boldsymbol{PP} \mid \boldsymbol{C}, \boldsymbol{P}, \boldsymbol{A}, \boldsymbol{\Theta}) \cdot \mathbb{P}(\boldsymbol{C_p} \mid \boldsymbol{C_m}, \boldsymbol{P}, \boldsymbol{A}, \boldsymbol{\Theta})$$
$$\cdot \mathbb{P}(\boldsymbol{C_m} \mid \boldsymbol{P}, \boldsymbol{A}, \boldsymbol{\Theta}) \cdot \mathbb{P}(\boldsymbol{\psi}) \cdot \mathbb{P}(\boldsymbol{P} \mid \boldsymbol{\omega}) \cdot \mathbb{P}(\boldsymbol{\omega}). \tag{11}$$

Overall, taxa percentages and climate regions that better fit the constraints of the prior climate and proxy pool modules should be weighted higher. How this is done in detail is described in the following.

In the context of MCMC sampling, we update $\mathbb{P}(\boldsymbol{P} \mid \boldsymbol{\omega})$ using the random walk Metropolis–Hastings (rwMH) technique, since a corresponding full conditional $\mathbb{P}(\boldsymbol{\omega} \mid \boldsymbol{P})$ does not follow a probability distribution from which we can sample directly. Without further prior information, we assume a uniform distribution across all taxa K at the beginning of the MCMC simulation:

$$\mathbb{P}(\boldsymbol{P} \mid \boldsymbol{\omega}) = \mathrm{Unif}(1, ..., \mathrm{K}). \tag{12}$$

The respective weights are determined with the help of an additional prior distribution:

$$\mathbb{P}(\boldsymbol{\omega}) = \mathrm{Dir}(\omega_1, ..., \omega_\mathrm{K} \mid \frac{1}{2}, ..., \frac{1}{2}). \tag{13}$$

Such a Dirichlet distribution allows us to determine the taxa weights as we have requested above. This means that the taxa weights have values between zero and one and add up to one. The Jeffreys prior hyperparameters $\frac{1}{2}$ of this distribution give each taxon equal prior weight. Furthermore, these values provide a weaker constraint for determining the posterior taxa weights.
This property follows directly from the characteristics of the Jeffreys prior (Gelman et al., 2013).

As described above, we want to sample not only taxa weights but also climate values in the climate feature space of the biome transfer function. In this way, we can identify preferred climate ranges based on the plant data and boundary conditions. The parameters $\boldsymbol{\psi}$ remain unchanged because we assume that they are a good approximation for the Holocene. Instead, we sample directly from the climate space and use $\mathbb{P}(\boldsymbol{C} \mid \boldsymbol{\psi})$ from Eq. 3. Again, rwMH is used because we cannot sample directly
from the full conditional. In this case, we use double truncated normal distributions $\mathcal{N}_t$ restricted to the climate range of the transfer function as proposal distributions to exclude biologically unrealistic climate values:

$$\mathbb{P}(\boldsymbol{C} \mid \boldsymbol{\psi}) = \mathcal{N}_t(\boldsymbol{C} \mid \boldsymbol{\mu}(\boldsymbol{\psi}), \boldsymbol{\sigma}(\boldsymbol{\psi}), \boldsymbol{a}(\boldsymbol{\psi}), \boldsymbol{b}(\boldsymbol{\psi})). \tag{14}$$



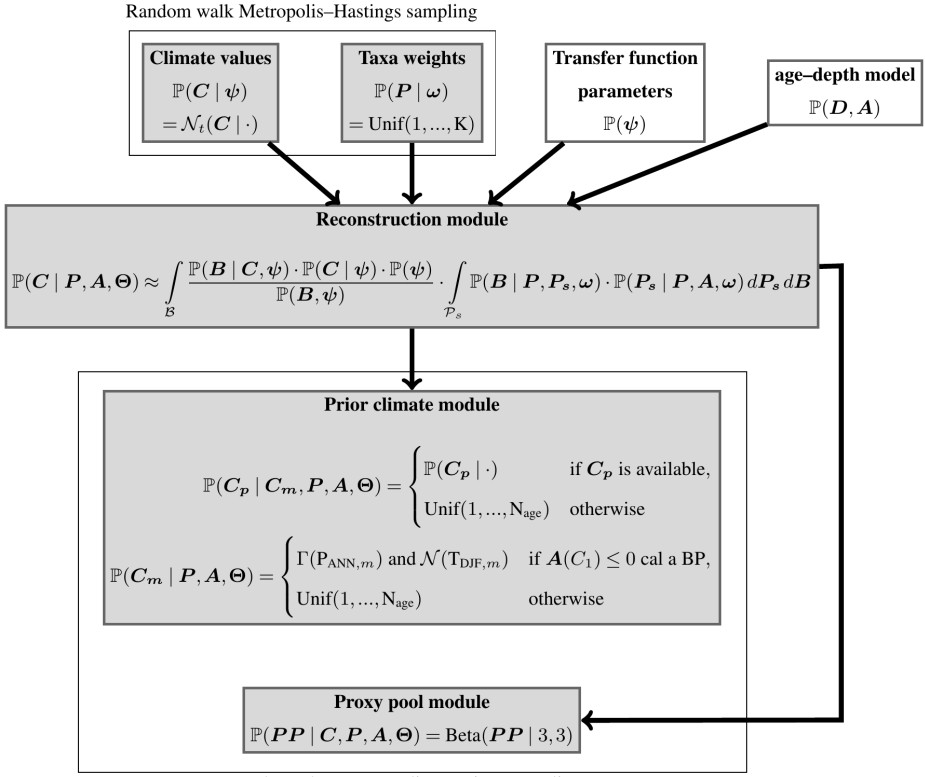

**Figure 3.** Directed acyclic graph of the Bayesian framework in Eq. 11. The gray boxes represent the quantities that will be inferred during MCMC sampling, and the white boxes contain fixed quantities. The corresponding arrows represent the mutual dependencies, with their direction pointing to the ascending hierarchical levels and the additional boxes indicate the respective sampling procedures of the modules contained therein.

The transfer function parameters $\psi$ determine not only the truncation ranges $a$ and $b$ but also the expectation values $\mu$ and standard deviations $\sigma$.

Fig. 3 summarizes graphically how this local reconstruction framework works. The boxes in the upper row contain the input variables, while those in the white boxes are not inferred during the MCMC simulation. The parameters of the transfer functions are defined in the next Sect. 3.5 and the age–depth relationship is described in Miebach et al. (2022). The upper gray boxes describing the inference of the taxa weights and the climate values via rwMH sampling. This is done by comparing the sampled climate reconstructions (reconstruction module) with additional recent climate data and an AP reference curve (prior climate

and proxy pool modules) and constraining them accordingly. These comparisons are made using the independent Metropolis–Hastings sampling. All in all, the procedure presented here not only avoids disadvantages two to six from the introduction, but also offers comprehensive extensions.



## 3.5 Transfer functions

One objective of this work is to systematically test a variety of possible methods to determine the transfer function $\mathbb{P}(\boldsymbol{B} \mid \boldsymbol{C}, \boldsymbol{\psi})$

from Eq. 3 and select the most appropriate algorithm for the task at hand. For this purpose we use the R package caret (Kuhn et al., 2019). This stands for **c**lassification **a**nd **re**gression **t**echniques and provides a variety of models that can be used to solve corresponding problems. The package supplies a simple way to compare the selected models via cross-validation. In this process, the provided data (cf. Fig. 1 (d)) are split into a training and a validation dataset. Cross-validation is performed on the training set (James et al., 2013), which accounts for $70\,\%$ of all data. Statistical verification distributions result from this,

which are used to derive the performance of the models. Cross-validation is also performed for a certain number of different parameters for the respective machine learning (ML) algorithms (model tuning). The entire process is very easily accessible in caret and runs completely automatically after the initial parameters have been defined. The remaining $30\,\%$ (hold-out set) are used to validate the models obtained by cross-validation on the remaining $70\,\%$. This has the advantage that they can be tested on an independent data set, further minimizing the risk of overfitting.

As can be seen in Fig. 1 (d), the defined biomes (minority classes) and the unspecified biome (majority class) are unbalanced. This means that the number of gridpoints covering the different classes varies greatly. In a balanced data set, they would be roughly equal. One could reduce the size of the entire map section so that the groups are more balanced. However, the models then deliver significantly worse and sometimes more unrealistic results. This problem is discussed for example in Thoma (2017) or in Weitzel et al. (2019). Thus, a model could provide higher probabilities of occurrence, on the one hand, where the

biomes does not occur in the feature space and, on the other hand, where the climate values are biologically unrealistic. When the map section is enlarged, this problem recedes, especially if the absence values can serve as a boundary. This is the case when the occurrence domain is enclosed by the absence domain in the two dimensional feature space spanned by $T_{DJF}$ and $P_{ANN}$. The reduction of the map section is analogous to the techniques of random under-sampling (Hoens and Chawla, 2013). The majority class is randomly reduced to the size of the minority class, potentially losing important information. In contrast,

random oversampling of the minority class risks overfitting. To solve this problem, the **S**ynthetic **M**inority **O**versampling **Te**chnique (SMOTE) is used (Bowyer et al., 2011). Here, a minority class instance is first randomly selected and its k-nearest minority class neighbors are determined. A line segment is then formed between one randomly selected neighbor in feature space. A synthetic instance of the minority class is created by selecting a random point along this line (Hoens and Chawla, 2013). SMOTE can only do this with one minority class at a time. Therefore, we use this technique separately for each of the

three minority classes compared to the majority class. Finally, all four classes consist of a similar number of data points. These are the input for the calculation of the transfer function in the ML competition. So only the training data are processed with SMOTE. For the model verification on the hold-out set, the original data are used.

Table 2 lists four ML models that we compete against each other. We have removed Support Vector Machines (SVMs) from this list as they are not competitive due to their disproportionately long prediction time. Similar difficulties with SVMs are also

found in Jergensen et al. (2020), where a ML competition for forecast models of convective storms is presented.





**Table 2.** Machine learning algorithms which are used for the competition:

| Algorithm: | Shortcut: | Citation: |
| --- | --- | --- |
| Artificial Neural Networks | NNET | Venables and Ripley (2002a) |
| Quadratic Discriminant Analysis | QDA | Venables and Ripley (2002b) |
| Mixture Discriminant Analysis | MDA | Leisch et al. (2017) |
| Gradient Boosting Machines | GBM | Greenwell et al. (2019) |

Comparatively simple classification problems arise in this work, so relatively simple artificial neural network structures (ANN) can be used. These deliver similarly good results with significantly less computational cost and the risk of overfitting is generally lower with simpler structures. After initial tests, the ANN from the nnet package is chosen in this work (NNET). It is a feedforward neural network that allows one hidden layer with an arbitrary number of hidden neurons (Venables and Ripley, 2002a).

Discriminant analysis involves the development of discriminants, i.e. linear combinations of independent variables that discriminate the categories of the dependent variable (James et al., 2013). QDA, for example, extracts discriminants that maximize separation between groups and then uses them to perform a Gaussian classification. QDA accounts for heterogeneity in the covariance matrices of these groups. Mixture Discriminant Analysis (MDA) can be considered as an extension that modifies the within-group multivariate density of predictors by a mixture (i.e., a weighted sum) of multivariate normal distributions (Rausch and Kelley, 2009).

Gradient Boosting Machines (GBM) is chosen to introduce an ML algorithm based on decision trees. It is a generalization of tree-boosting that attempts to mitigate the following problems: Speed, interpretability, and robustness to overlapping group distributions and, most importantly, mislabeling of the training data (Hastie et al., 2009). Thus, it creates an accurate and effective standard procedure.

The approach presented here to systematically identify the most appropriate method to describe the relationship between botanical data and climate remedies the last disadvantage mentioned in the introduction.

## 4 Results

This section first presents the results of the machine learning competition. Afterwards, the reconstruction of Lake Kinneret and the corresponding MCMC data are shown.



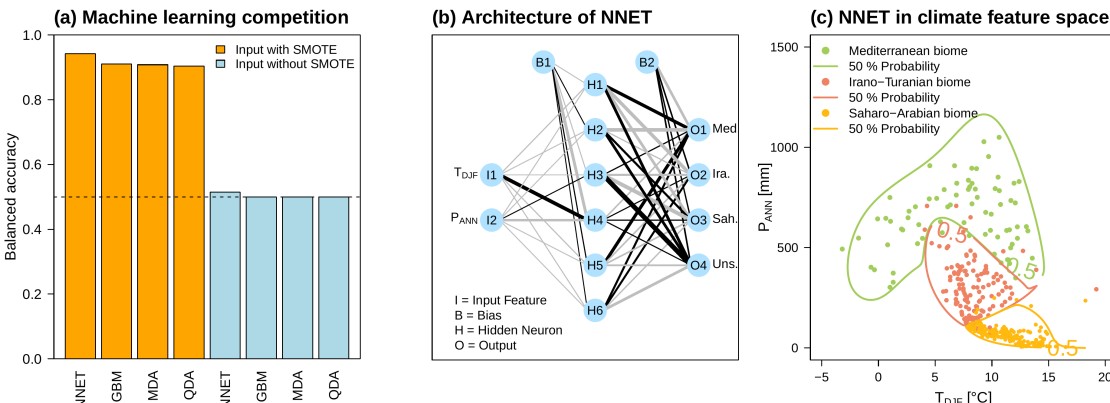

**Figure 4.** Panel (a) summarizes the balanced accuracy of all ML algorithms based on the original input data and the data modified with SMOTE. The winner of this ML competition is the feedforward neural network shown in (b). The thickness of the respective lines reflects the relative absolute value of the parameters. Furthermore, the gray lines stand for negative values and the black lines for positive values. In panel (c) the classification in the feature space of $T_{DJF}$ and $P_{ANN}$ is shown: the colored solid lines represent the $50\%$ probability of the biomes occurring based on the transfer function from (b). The corresponding original input data are also shown as colored dots.

### 4.1 Machine learning competition

In the following, the results of the machine learning competition are analyzed in detail. The evaluation focuses on the problem of unbalanced data sets. These are augmented with SMOTE until the input values are balanced. Subsequently, the models are trained on these data sets and finally evaluated with a fraction of the original data.

In our work, this classification is based on the so-called balanced accuracy (BA), which is calculated using $2 \times 2$ contingency tables of predicted data compared to hold-out validation data. From these, the true positive and true negative rates can be calculated, referred to as sensitivity and specificity, respectively (Chicco et al., 2021). The arithmetic mean of these two measures is the BA, which is an appropriate metric for trained ML models designed to describe an unbalanced data set (Brodersen et al., 2010). BA varies between zero and one, with values close to one indicating well-performing classifications.

The results of all trained models are shown in Fig. 4 (a). A distinction is made between models trained on the original data set (without SMOTE) and those trained on data augmented with SMOTE. It is immediately noticeable that the results marked by the blue boxplots have a BA of 0.5 (except NNET). In these instances, the sensitivity is always zero and the specificity is one, which means that no presence is predicted. In contrast, the other fits (orange boxplots) have an average BA of about 0.92, which is a significant increase. Thus, we can not only obtain fitted models with high significance, but also reduce the boundary

effects in the feature space, resulting in more closed contour lines as shown in Fig. 4 (c). Although all algorithms provide good trained models on their own (cf. Fig. 4 (a)), the direct comparison between them leads to the result that a simple artificial neural network emerges as the winner. The structure of this NNET is shown in Fig. 4 (b), where the two climate variables represent the input layer and the three biomes with the unspecified biome the output layer. Furthermore, six hidden neurons proved to



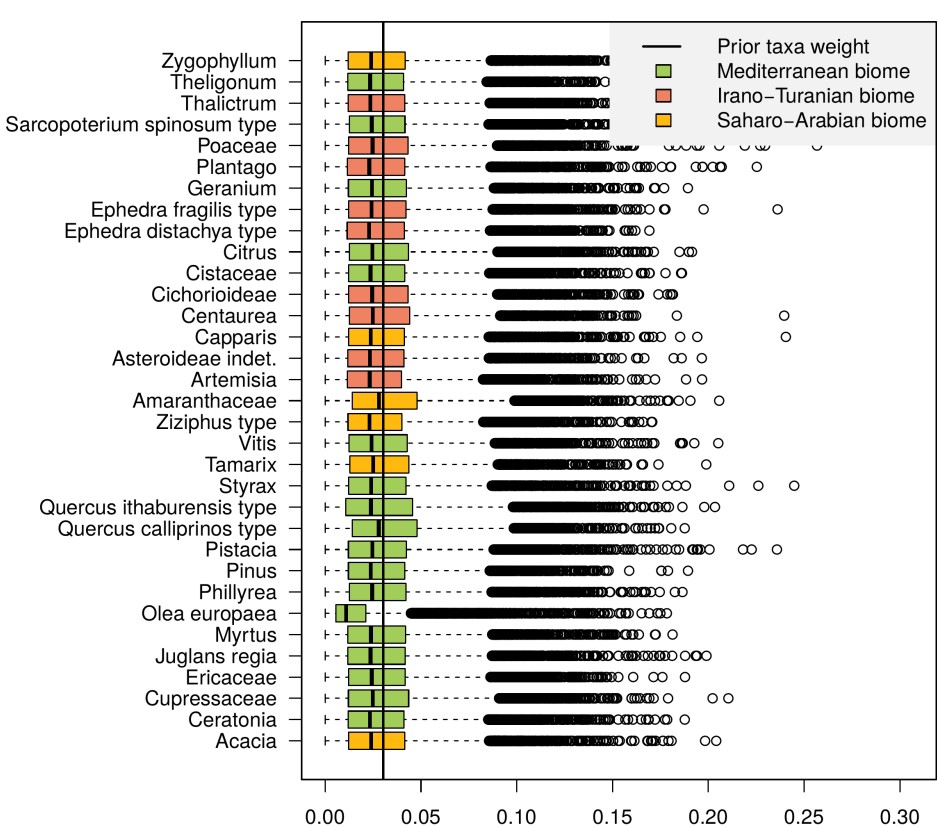

**Figure 5.** Posterior and prior taxa weights. The solid black line indicates the prior and the boxplots the posterior taxa weights. In addition, the assignment of each taxon to the three biomes is color-coded.

be the best compromise between BA and overfitting in model tuning. This network structure is finally used for the following
climate reconstruction.

## 4.2 Quantitative reconstruction

Due to the large number of parameters, we decide to generate 1 million MCMC samples. To make such a reconstruction as fast as possible, C++ is used. Thus, a reconstruction on a standard CPU takes only about 40 seconds. Finally, the first quarter of the sample size is considered burn-in and every 75th iteration of the remaining samples is included to avoid the autocorrelation in
the MCMC procedure.



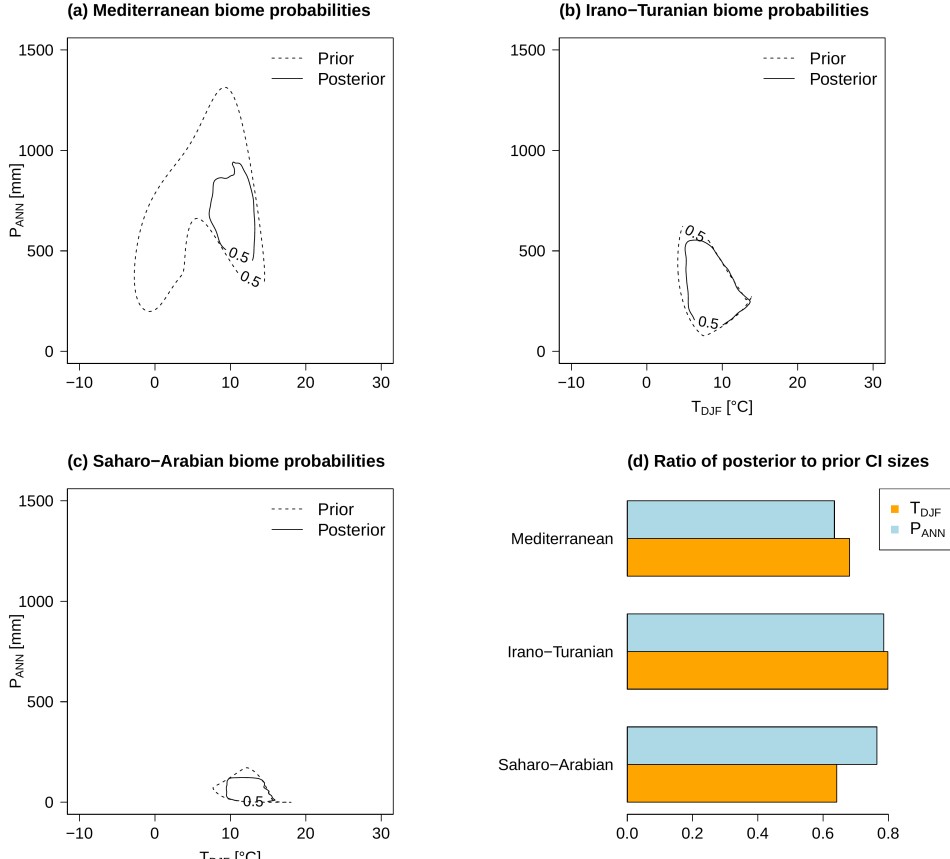

**Figure 6.** In panels (a)–(c), the prior biome probabilities of $50\,\%$ are indicated with dashed black lines and the corresponding posterior biome probabilities are depicted with solid black lines. In (d), the ratios of the $95\,\%$ credible interval (CI) of the corresponding prior and posterior distributions from panels (a)–(c) are shown.

First, the stochastic behaviour of this MCMC simulation must be tested for convergence. For this, we use the multivariate extension of the Gelman–Rubin convergence indicator (Brooks and Gelman, 1998). The closer this is to one, the more likely it is that convergence has been achieved. Gelman et al. (2013) recommend a value of less than 1.1. In our case, this is 1.001, from which we conclude that this simulation setup converges.

335 Fig. 5 summarizes the posterior taxa weights $\mathbb{P}(\boldsymbol{P}\mid\boldsymbol{\omega})$ determined by this simulation in boxplots. It is immediately apparent that, with the exception of *Olea europaea, Quercus calliprinos*, and Amaranthaceae, the mean posterior taxa weights deviate only slightly from the prior uniform distribution. In particular, the olive taxon receives a considerably lower weight, which is due to the generally high pollen percentage in the core (see Appendix B for details). To ensure a sufficiently high variability with respect to the reference curve of AP in Fig. 2 (b), the new reconstruction method weights *Quercus calliprinos* and

340 Amaranthaceae highest.



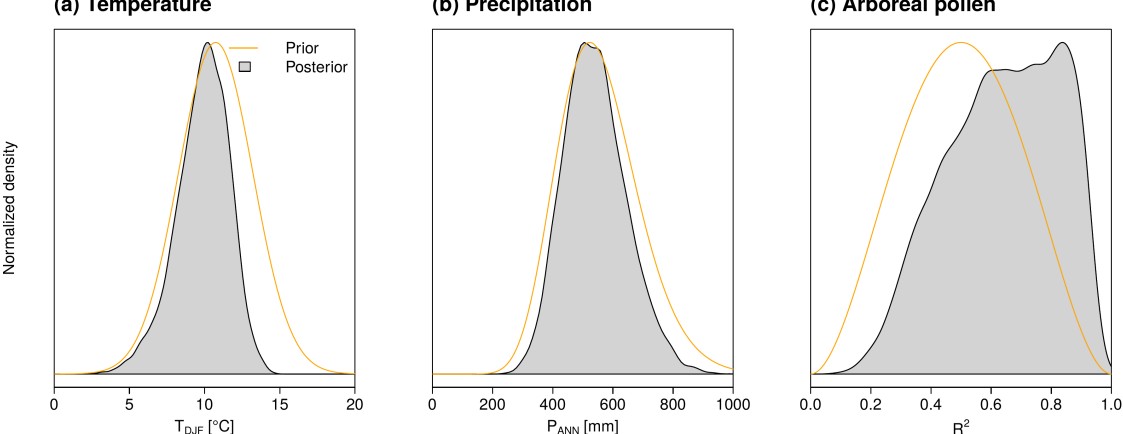

**Figure 7.** The prior proposal distributions (orange lines) and the posterior samples (gray areas) of $T_{DJF}$ in (a), $P_{ANN}$ in (b) and the explained variance $R^2$ of the reconstructions compared to the arboreal pollen reference curve in (c).

Fig. 6 shows the prior and posterior probability distributions $\mathbb{P}(\boldsymbol{B} \mid \boldsymbol{C}, \boldsymbol{\psi})$. We see the largest changes within the Mediterranean biome in (a). The branch with lower temperature and precipitation of this distribution leads to reconstructions that cannot fulfil the boundary conditions. The corresponding posterior probability reveals an average $T_{DJF}$ of $10\,°C$ and an $P_{ANN}$ of $700\,\text{mm}$. In panel (d) we see the reduced posterior variances of the climate variables within the biomes. The temperature

distribution of the Saharo-Arabian biome, for example, must be constrained so that it does not contradict the boundary conditions of the most recent temperature data. Overall, it can be seen that the posterior temperatures settle at around $10\,°C$ and thus show less variability than the corresponding precipitation distribution.

The posterior samples described above are determined with the prior boundary conditions in Fig. 7. We also see the corresponding posterior distributions as gray areas. It is noticeable that the temperature in (a) and the precipitation in (b) have

slightly lower values. In contrast, relatively high explained variances are reconstructed in (c). Based on the taxa weights and the values of the transfer functions from Fig. 5 and Fig. 6, it can be concluded that a trade-off with respect to recent climate conditions is reached when the median $R^2$ is around 0.65 ($50\,\%$ CI from 0.50 to 0.80).

In the following, we describe the final reconstruction. It is divided into the percentages of the reconstructed biomes $\mathbb{P}(\boldsymbol{B} \mid \boldsymbol{P}, \boldsymbol{A}, \boldsymbol{\omega})$ in Fig. 8 and the reconstructed $T_{DJF}$ and $P_{ANN}$ in Fig. 9. From the former, we can infer the importance of these biomes

in specific periods.

The period **9–7 cal ka BP** can be associated mainly with the Pottery Neolithic. The vegetation is described in Schiebel and Litt (2018) with a strong influence of steppe vegetation in the catchment area of LK. They conclude that this is due to increasing drought, which is confirmed by the increased percentages of the Saharo-Arabian and Irano-Turanian biomes. In contrast, the Mediterranean biome records comparatively low percentages during this period. This leads, on the one hand, to

the highest average $T_{DJF}$ of over $10\,°C$ and, on the other hand, to relatively low $P_{ANN}$ of about $400\,\text{mm}$. Furthermore, Miebach




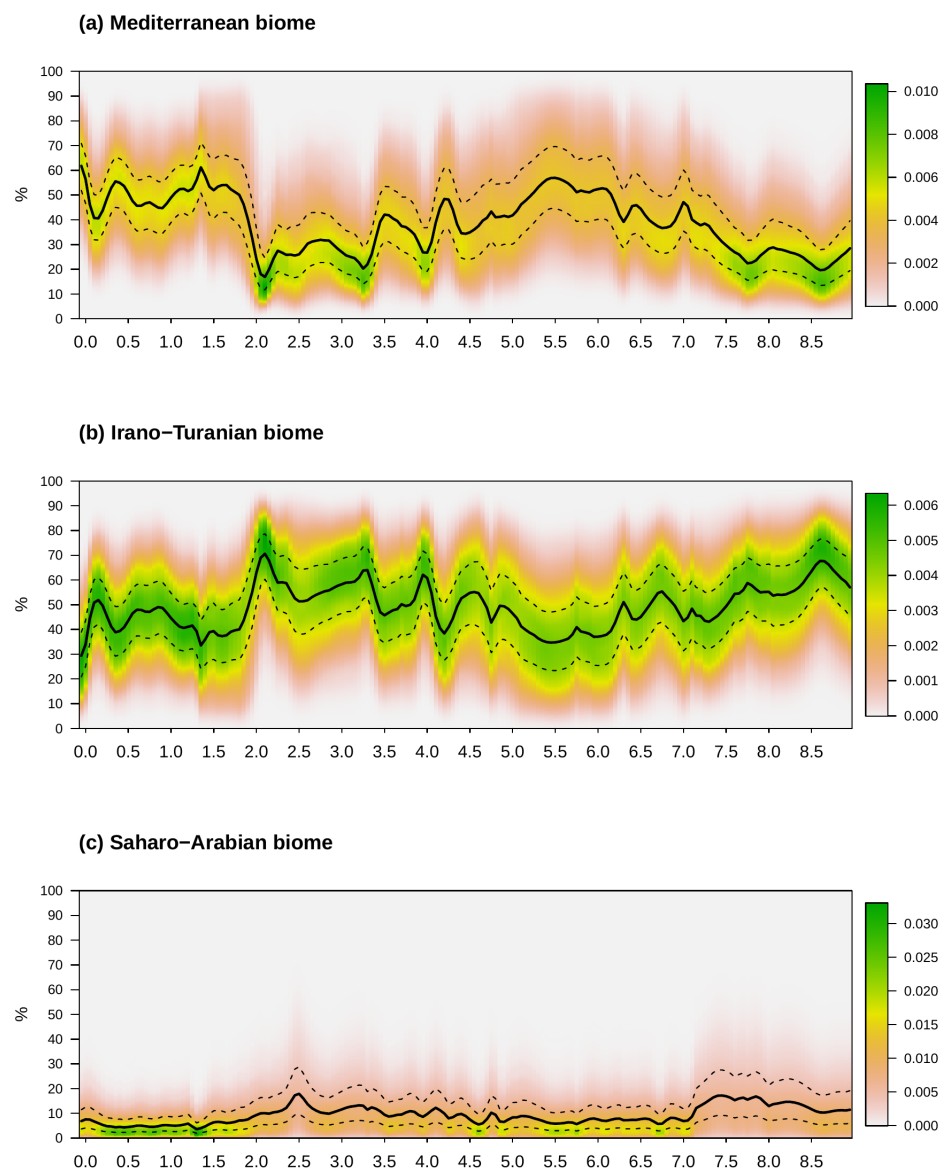

**Figure 8.** Posterior biome percentages in relation to cal ka BP. The colors indicate the probability density values, the black solid lines its median, and the dashed black lines the first and third quartiles.

et al. (2022) infers a weak cooling trend and precipitation decrease during 7.8–6.6 cal ka BP from carbon isotope signals of the Sea of Galilee. These qualitative statements are confirmed by the new climate reconstruction within both variables.

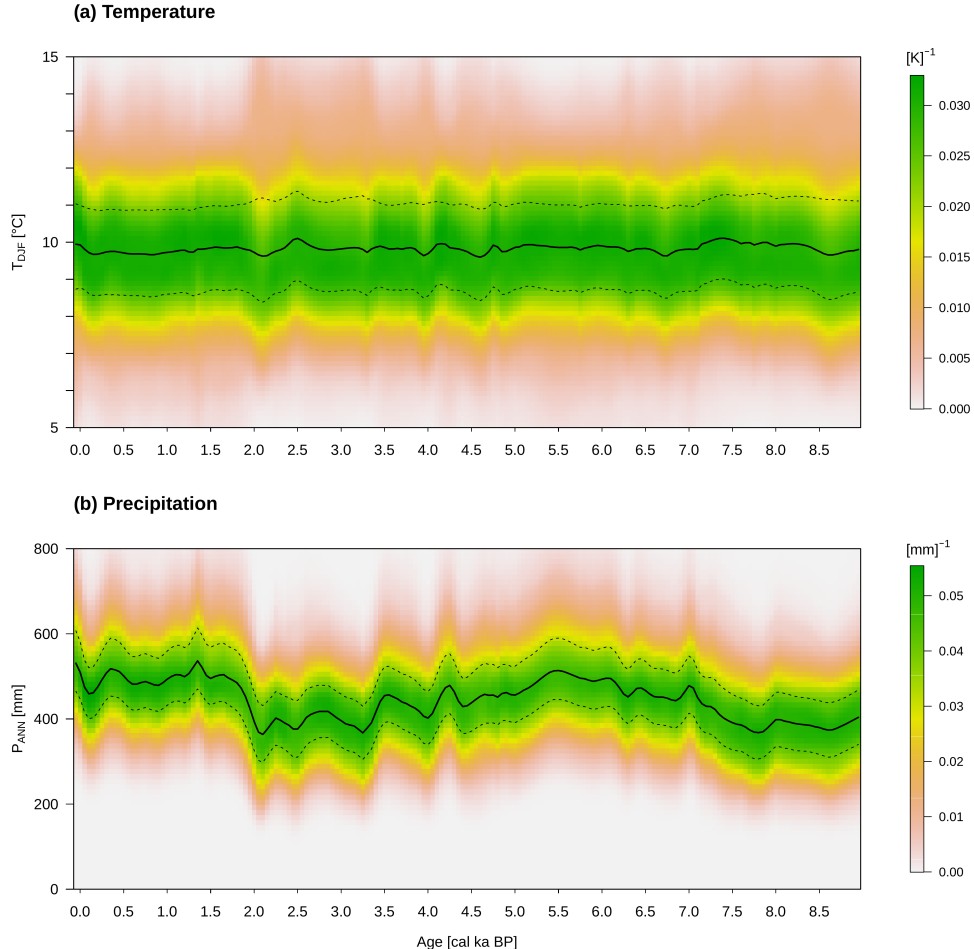

**Figure 9.** As Fig. 8, but for the quantitative paleoclimate reconstruction of the Lake Kinneret region. In (a) the reconstructed $T_{DJF}$ in °C and in (b) the $P_{ANN}$ in mm are shown.

The beginning of the period **7–5 cal ka BP** is accompanied by an increase in *Olea europaea* and thus, the Mediterranean biome. Schiebel and Litt (2018) assume climate change towards higher precipitation, which is also confirmed by our recon-

struction. The average $P_{ANN}$ is about $500\,\mathrm{mm}$ and temperatures surrounding $10\,°\mathrm{C}$. During the Chalcolithic (ca. 6.5–5.5 cal ka BP), precipitation shows a local maximum, which decreases after about 5.5 cal ka BP. Such behaviour could be related to the transition from the Chalcolithic to the Early Bronze Age.

The Early Bronze Age to Iron Age within **5–2.3 cal ka BP** reflects not only human-induced but also climatically driven vegetation changes. On the one hand, Schiebel and Litt (2018) describe the end of olive cultivation around 5 cal ka BP as a

human influence. On the other hand, the decrease in oak pollen of 4 and 3.2 cal ka BP could be related to the Bond events of 4.2 and 3.2 associated with droughts in the Levant. During this period, the precipitation shows a steady decline from ca.





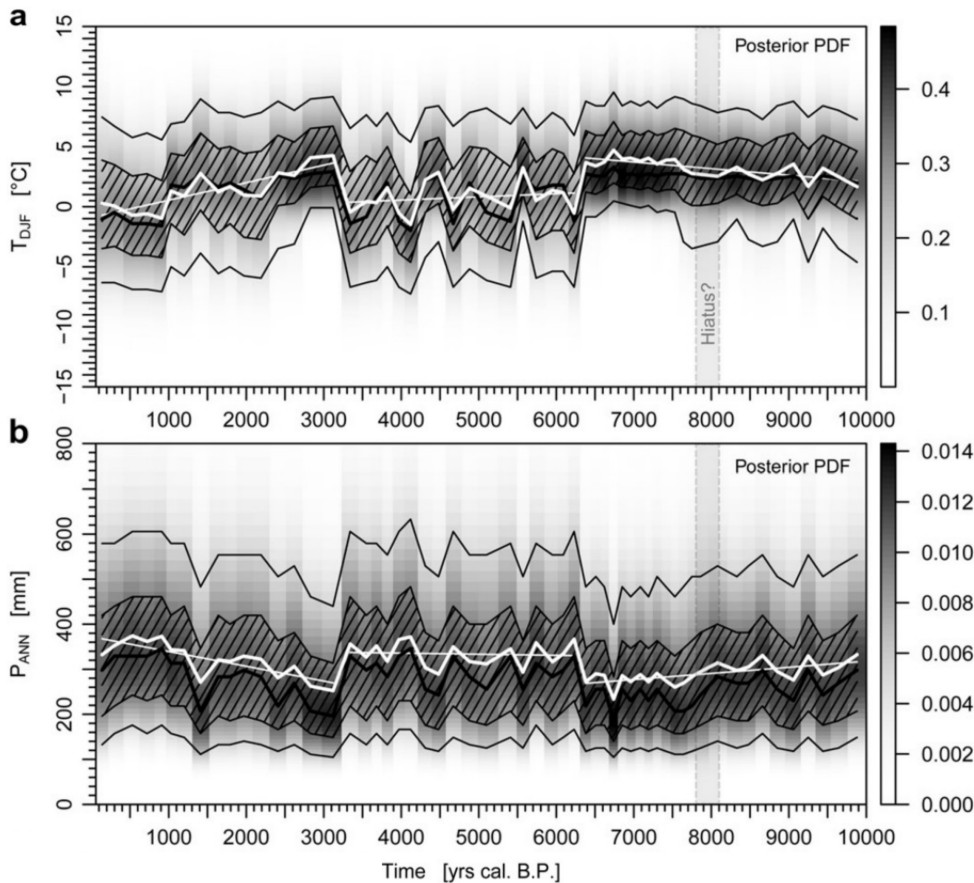

**Figure 10.** Paleoclimate reconstruction of the Dead Sea, modified after Litt et al. (2012). In (a) the $T_{DJF}$ anomaly in °C and in (b) the $P_{ANN}$ in mm are shown. The thicker white lines are the expectation values and the thinner white lines describe the respective linear climate trends. The thicker black lines mark the mode and the thinner black lines indicate the interdecile and interquartile ranges.

500 mm to about 400 mm, while the temperature remains around 10 °C. At the same time, the Mediterranean biome decreases and the others increase. The climate change to lower precipitation around 4 cal ka BP could be related to the transition from the Early to the Middle Bronze Age. The second and larger variation during 3.2 cal ka BP might be related to the collapse of

the Late Bronze Age (Langgut et al., 2013). Furthermore, the Iron Age in the Near East lasted from about 3.1 - 2.5 cal ka BP (Langgut et al., 2013). This corresponds to an increase in precipitation at the beginning and ends in a minimum with values around 400 mm. 2.5 cal ka BP marks the transition from the Iron Age to the Babylonian–Persian period, which lasted about 200 years and is accompanied by a slight increase in precipitation. A decrease in olive pollen around 5 cal ka BP cannot be associated with changed climatic conditions (Schiebel and Litt, 2018) and is also not visible in the reconstruction. This is made

possible by the lower weighting of this taxon and supports the choice of the beta distribution parameters of $R^2$.



The years from **2.3–1.5 cal ka BP** are marked by the Hellenistic and Roman–Byzantine periods. This can be associated with the Roman Climatic Optimum (Langgut et al., 2013) and a noticeable increase in precipitation can be seen in the reconstruction. Orland et al. (2009) recognize from isotopic data from Soreq Cave a decrease in precipitation during the period 1.9–1.3 cal ka BP. They suggest that this climate change weakened the economic system of the Roman and Byzantine Empires, which
contributed to the decline of their rule in the Levant.

This leads us to the early Islamic period to the present from **1.5–0 cal ka BP**. The reconstructed $P_{ANN}$ shows relatively high values and exhibits only minor variations. Finally, the climate PDFs of the youngest timeslice are the same as the posterior distributions depicted in Fig. 7 (a) and (b).

In comparison with the quantitative climate reconstruction of the Dead Sea in Fig. 10, we can observe some similarities.
During the early Holocene, relatively low precipitation is reconstructed up to 6.5 cal ka BP. These increase markedly during the mid-Holocene up to 3.3 cal ka BP. They then fall significantly and rise in the further course until the youngest time slice. With the corresponding $T_{DJF}$, the trend is exactly the opposite. Overall, we see similar patterns, although the temperature fluctuations in Litt et al. (2012) are larger, which is due to the special location of the Dead Sea as a transition zone of the three biomes.

## 5 Discussion and possible extensions

Our new approach of a local climate reconstruction offers a systematic method to investigate the variability of the data under certain boundary conditions. These partly originate from sources other than the original botanical proxy data. In this way, it can be determined whether a physically and biologically realistic climate reconstruction is possible with the given proxy data. The new method shows, for example, that the probability of the Mediterranean biome with lower temperatures and precipitation in Fig. 6 (a) cannot be used when constrained by recent climate data and arboreal pollen reference. So far, the full distributions
have been included in the reconstructions. This new flexibility in terms of transfer functions accounts for the assumption that the relationship between recent biome distributions and the corresponding climate remains unchanged in space and time. The posterior distributions in Fig. 8 show where these might have been on average during the reconstruction period for the Sea of Galilee.

Further useful information can be obtained from the posterior taxa weights. From this, it can be deduced to what extent
a particular taxon is included in the reconstruction based on its occurrence in the sediment core. Thus, this automatically determined data can expand the underlying expert knowledge. Here it seems that less weight needs to be given to olive pollen, which dominates at certain depths, to ensure the influence of the other taxa. This shows how the highest possible variability can be obtained from the proxy information under the assumed boundary conditions. With a comparatively higher weighting of *Quercus calliprinos*, the recent precipitation distribution at the Sea of Galilee can be approximated as well as possible.
Furthermore, we find the highest weights in relation to the Irano-Turanian and Saharo-Arabian biomes in the taxa Poaceae and Amaranthaceae. This makes it possible to reconstruct the lower precipitation in the periods 9–7 cal ka BP and 3.2–2 cal ka BP. It also helps to reduce the human impact on vegetation during the reconstruction process. This is particularly striking in the Mediterranean biome around 5 cal ka BP, where we see only minor changes.





In the posterior distribution of the explained variance between the reconstructed precipitation and AP in Fig. 7 (c), values
of 0.65 occur on average. These relatively high positive correlations confirm the relationship between these two variables
proposed in Schiebel and Litt (2018). We also see the order of magnitude in which this must be present to allow a compromise
with the other boundary conditions in Fig. 7 (a) and (b).

Compared to previous local climate reconstructions based on Bayesian statistics, the proxy information considered can be
included without further processing. This means that it is not necessary to pre-select specific plant data and set thresholds for
their probability of occurrence. In addition, the boundary conditions such as climate anchor points and reference curves can be
extended. For example, isotope data from the Mediterranean Sea such as Medstack (Colleoni et al., 2012) or from speleothems
in the Soreq Cave (Bar-Matthews et al., 2003) can be used as guidelines. In addition, PDFs for the MH from paleoclimate
simulations (Braconnot et al., 2011) can be included. The new reconstruction method can therefore be easily adapted and used
accordingly in future studies.

The age uncertainty accounted for in this study with the new age–depth transformation presented allows for data-driven
smoothing along with stretching/compressing of the original depth axis of the proxy information, as well as arbitrarily high
resolution and a regular temporal grid. This means that reference proxies can now be examined in spectral space. For example,
the fluctuations around 4 and 3.2 cal ka BP could be compared with the ice rafted debris of the North Atlantic using wavelet
power spectra (Debret et al., 2007). We thus see that the reconstruction method presented can be extended with additional
independent proxy information, so that quantitative multiproxy analyses are possible as well as the inclusion of results from
paleoclimate simulations.

## 6   Conclusions

In this study, we present new techniques for generating local paleoclimate reconstructions based on botanical proxies. For this
purpose, we use a newly developed BHM solved with MCMC sampling. To place the proxy information in a temporal context,
a new probabilistic method is used to assign age information to depths in sediment cores. In particular, the uncertainty of age
is accounted for by a separate BHM introduced in this work. Climate variables such as $T_{DJF}$ and $P_{ANN}$ were included using
a transfer function based on biome occurrence. We determine these functions with a machine learning competition. Such a
systematic identification of the most appropriate method to describe the relationship between botanical data and climate is
performed here for the first time.

These new techniques are applied to plant data from the Sea of Galilee during the Holocene. The reconstructed climate
variables reflect the qualitative climate reconstructions explained in Schiebel and Litt (2018); Miebach et al. (2022); Orland
et al. (2009). Moreover, the algorithm is able to find climate changes that can be associated with Bond events and known
archeological and cultural changes in the Levant. Furthermore, there is a connection with the quantitative reconstruction of the
Dead Sea in Litt et al. (2012), where similar climatological trends are reconstructed.

Overall, our new methods provide an additional way to calculate quantitative paleoclimate reconstructions. From our results,
we conclude that more automatic, statistics-based techniques complement those that require additional assumptions. Further-



more, our model provide additional information such as taxa weights and biome climate ranges with corresponding uncertainty estimates. From this, we can gain new insights into possible biological mechanisms involved in ecological changes caused by past climate variability. The new methods not only remedies all the disadvantages mentioned in the introduction, but also

represents an attempt to solve complex BHMs with little computational cost. Extending this to multiple proxy sources and applying it to other geographical areas could qualitatively and quantitatively expand knowledge about the climate history of the Earth.

*Code and data availability.* There are two Zenodo repositories, written in R (Netzel, 2023b) and in Python (Netzel, 2023c). These each include the Bayesian framework with the MCMC simulation in C++. The required input data from the sediment core, from the ML competition

and from the age–depth model are available in the corresponding repositories.





## Appendix A: Derivation of the local reconstruction model

Using Bayes' theorem, we can express the probability distribution for climate $\boldsymbol{C}$ given pollen and macrofossils $\boldsymbol{P}$, depth $\boldsymbol{D}$, and parameter $\boldsymbol{\Theta}$. In the process, we also introduce the biome information $\boldsymbol{B}$:

$$\mathbb{P}(\boldsymbol{C} \mid \boldsymbol{P}, \boldsymbol{D}, \boldsymbol{\Theta}) = \int_{\mathcal{B}} \mathbb{P}(\boldsymbol{C} \mid \boldsymbol{P}, \boldsymbol{B}, \boldsymbol{D}, \boldsymbol{\Theta}) \cdot \mathbb{P}(\boldsymbol{B} \mid \boldsymbol{P}, \boldsymbol{D}, \boldsymbol{\Theta}) \, d\boldsymbol{B}. \tag{A1}$$

In the case of a finite number of taxa, the integral is a corresponding sum. Consider $\mathbb{P}(\boldsymbol{C} \mid \boldsymbol{P}, \boldsymbol{B}, \boldsymbol{D}, \boldsymbol{\Theta})$ in more detail:

$$\mathbb{P}(\boldsymbol{C} \mid \boldsymbol{P}, \boldsymbol{B}, \boldsymbol{D}, \boldsymbol{\Theta}) \overset{1.}{\approx} \mathbb{P}(\boldsymbol{C} \mid, \boldsymbol{B}, \boldsymbol{D}, \boldsymbol{\Theta}) \overset{2.}{\approx} \mathbb{P}(\boldsymbol{C} \mid \boldsymbol{B}, \boldsymbol{\Theta}) \overset{3.}{\approx} \mathbb{P}(\boldsymbol{C} \mid \boldsymbol{B}, \boldsymbol{\psi}) \overset{4.}{\approx} \frac{\mathbb{P}(\boldsymbol{B} \mid \boldsymbol{C}, \boldsymbol{\psi}) \cdot \mathbb{P}(\boldsymbol{C} \mid \boldsymbol{\psi}) \cdot \mathbb{P}(\boldsymbol{\psi})}{\mathbb{P}(\boldsymbol{B}, \boldsymbol{\psi})}. \tag{A2}$$

With the following assumptions and applications:

1. $\boldsymbol{C}$ is conditionally independent of $\boldsymbol{P}$ if $\boldsymbol{B}$ is given. This assumes that $\boldsymbol{B}$ explains enough variability of the core.

2. The link between $\boldsymbol{C}$ and $\boldsymbol{B}$ is conditionally independent of depth. This means that the relationship between these quantities is assumed to be unchanged for any depth and thus any age of the core. The assumption that this relationship has not changed over time is an important part of our reconstruction method. When we look at older time periods, we need to keep this in mind, as the relationship may well have changed due to evolutionary processes.

3. The connection between $\boldsymbol{C}$ and $\boldsymbol{B}$ is described only by the parameter $\boldsymbol{\psi}$. Furthermore, $\boldsymbol{\psi}$ and $\boldsymbol{\omega}$ are a priori independent: $\mathbb{P}(\boldsymbol{\Theta}) = \mathbb{P}(\boldsymbol{\psi}) \cdot \mathbb{P}(\boldsymbol{\omega})$.

4. Application of Bayes' theorem.

If we substitute Eq. A2 into Eq. A1, we get:

$$\mathbb{P}(\boldsymbol{C} \mid \boldsymbol{P}, \boldsymbol{D}, \boldsymbol{\Theta}) \approx \int_{\mathcal{B}} \frac{\mathbb{P}(\boldsymbol{B} \mid \boldsymbol{C}, \boldsymbol{\psi}) \cdot \mathbb{P}(\boldsymbol{C} \mid \boldsymbol{\psi}) \cdot \mathbb{P}(\boldsymbol{\psi})}{\mathbb{P}(\boldsymbol{B}, \boldsymbol{\psi})} \cdot \mathbb{P}(\boldsymbol{B} \mid \boldsymbol{P}, \boldsymbol{D}, \boldsymbol{\omega}) \, d\boldsymbol{B}. \tag{A3}$$

Eq. 2 allows us to transform this model from depth to age:

$$
\begin{aligned}
\mathbb{P}(\boldsymbol{C} \mid \boldsymbol{P}, \boldsymbol{A}, \boldsymbol{\Theta}) &= \int_{\mathcal{D}} \mathbb{P}(\boldsymbol{C} \mid \boldsymbol{P}, \boldsymbol{D}, \boldsymbol{\Theta}) \cdot \mathbb{P}(\boldsymbol{D} \mid \boldsymbol{A}) \, d\boldsymbol{D} \\
&\overset{\text{Eq.A3}}{\approx} \int_{\mathcal{D}} \int_{\mathcal{B}} \frac{\mathbb{P}(\boldsymbol{B} \mid \boldsymbol{C}, \boldsymbol{\psi}) \cdot \mathbb{P}(\boldsymbol{\psi})}{\mathbb{P}(\boldsymbol{B}, \boldsymbol{\psi})} \cdot \mathbb{P}(\boldsymbol{B} \mid \boldsymbol{P}, \boldsymbol{D}, \boldsymbol{\omega}) \, d\boldsymbol{B} \cdot \mathbb{P}(\boldsymbol{D} \mid \boldsymbol{A}) \, d\boldsymbol{D} \\
&\approx \int_{\mathcal{B}} \frac{\mathbb{P}(\boldsymbol{B} \mid \boldsymbol{C}, \boldsymbol{\psi}) \cdot \mathbb{P}(\boldsymbol{\psi})}{\mathbb{P}(\boldsymbol{B}, \boldsymbol{\psi})} \cdot \int_{\mathcal{D}} \mathbb{P}(\boldsymbol{B} \mid \boldsymbol{P}, \boldsymbol{D}, \boldsymbol{\omega}) \cdot \mathbb{P}(\boldsymbol{D} \mid \boldsymbol{A}) \, d\boldsymbol{D} \, d\boldsymbol{B} \\
&\overset{\text{Eq.2}}{\approx} \int_{\mathcal{B}} \frac{\mathbb{P}(\boldsymbol{B} \mid \boldsymbol{C}, \boldsymbol{\psi}) \cdot \mathbb{P}(\boldsymbol{\psi})}{\mathbb{P}(\boldsymbol{B}, \boldsymbol{\psi})} \cdot \mathbb{P}(\boldsymbol{B} \mid \boldsymbol{P}, \boldsymbol{A}, \boldsymbol{\omega}) \, d\boldsymbol{B}.
\end{aligned}
\tag{A4}
$$





Consider $\mathbb{P}(\boldsymbol{B} \mid \boldsymbol{P}, \boldsymbol{A}, \boldsymbol{\omega})$ in more detail:

$$\mathbb{P}(\boldsymbol{B} \mid \boldsymbol{P}, \boldsymbol{A}, \boldsymbol{\omega}) = \int\limits_{\mathcal{P}_s} \mathbb{P}(\boldsymbol{B} \mid \boldsymbol{P}, \boldsymbol{P_s}, \boldsymbol{A}, \boldsymbol{\omega}) \cdot \mathbb{P}(\boldsymbol{P_s} \mid \boldsymbol{P}, \boldsymbol{A}, \boldsymbol{\omega}) \, d\boldsymbol{P_s} \tag{A5}$$

The first term in the integral contains the information about which selected plant proxies $\boldsymbol{P_s}$ from the lake sediment belong to which biome. Note that a more detailed relationship in terms of age $\boldsymbol{A}$ could be added at this point. Because we consider the Holocene in this study, we assume that the probabilities for the biomes are conditionally independent of age if selected plant proxies are given. The second term describes the plant information from the sediment core in terms of age. Finally, we can substitute Eq. A5 into Eq. A4 and obtain the reformulated BBM:

$$\mathbb{P}(\boldsymbol{C} \mid \boldsymbol{P}, \boldsymbol{A}, \boldsymbol{\Theta}) \approx \int\limits_{\mathcal{B}} \frac{\mathbb{P}(\boldsymbol{B} \mid \boldsymbol{C}, \boldsymbol{\psi}) \cdot \mathbb{P}(\boldsymbol{C} \mid \boldsymbol{\psi}) \cdot \mathbb{P}(\boldsymbol{\psi})}{\mathbb{P}(\boldsymbol{B}, \boldsymbol{\psi})} \cdot \int\limits_{\mathcal{P}_s} \mathbb{P}(\boldsymbol{B} \mid \boldsymbol{P}, \boldsymbol{P_s}, \boldsymbol{\omega}) \cdot \mathbb{P}(\boldsymbol{P_s} \mid \boldsymbol{P}, \boldsymbol{A}, \boldsymbol{\omega}) \, d\boldsymbol{P_s} \, d\boldsymbol{B}. \tag{A6}$$

## Appendix B: Figures of the taxa spectrum

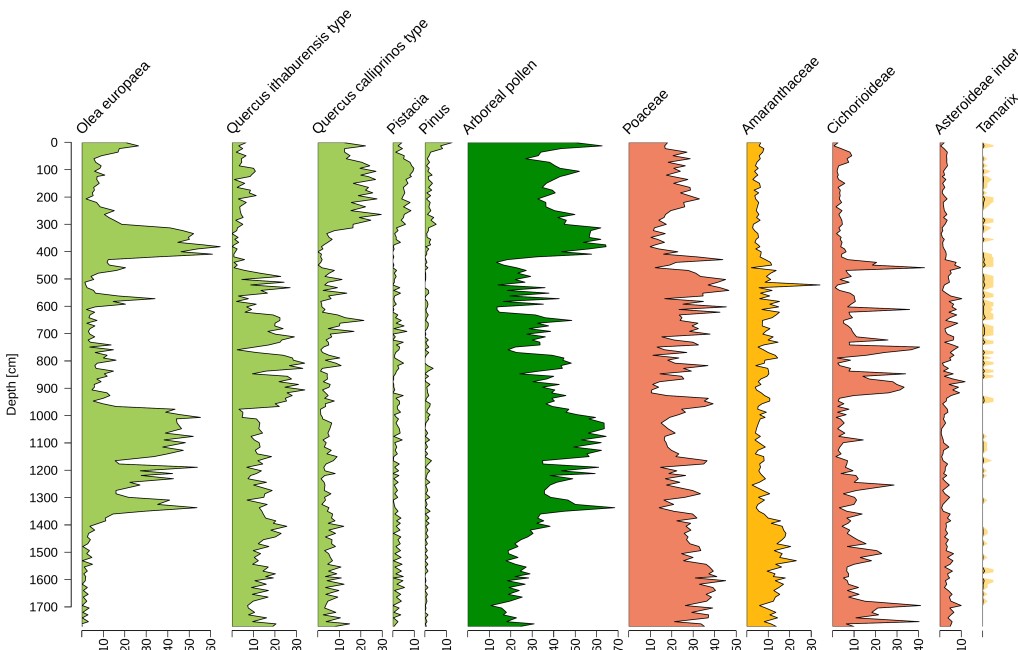

**Figure B1.** Percentage distribution of terrestrial pollen sums as a function of depth for some taxa from the Sea of Galilee. In the middle, the aggregated arboreal pollen are shown in dark green. The other colors correspond to the assignment to the respective biome.



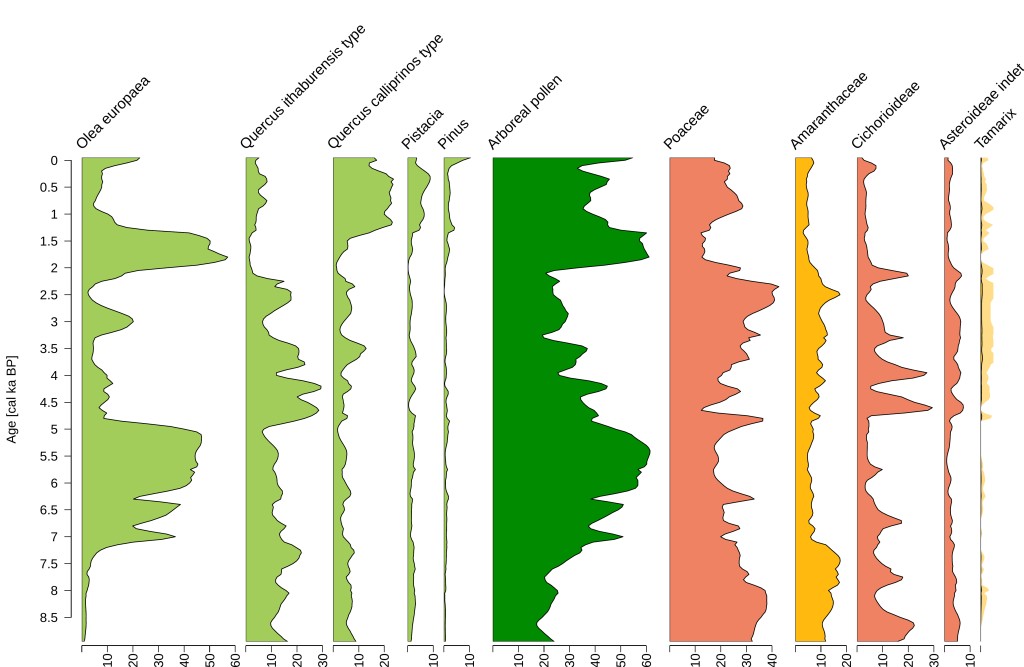

**Figure B2.** As Fig. B1, but with the new transformation from depth to age of sediment core.



*Author contributions.* TN did the theoretical work, implemented the model and graphics and wrote most of the text. AM wrote the sections Study Area, Material and Palynology. TL and AM provided the palynological data. AH and TL suggested the reconstruction framework, contributed to the discussion of the results and commented on the different versions of the paper.

*Competing interests.* The authors declare that they have no conflict of interest.

*Acknowledgements.* The basic idea and the palynological data for this work were elaborated within the framework of the Collaborative Research Centre (SFB) 806 "Our Way to Europe", which is funded by the German Research Foundation (DFG) [DFG project number 57444011]. We would like to thank Maarten Blaauw for reviewing the *Bacon.d.Age* function and including it in the rbacon package.



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
