# Peer review of "New probabilistic methods for quantitative climate reconstructions applied to palynological data from Lake Kinneret"

_EGUsphere, 2023_

## Author Response (AR1)

**General remark with respect to the Figures**

A new Fig 1 is added; Fig 6, Fig 7, Fig 8, Fig 9 and Fig 10 (in the first MS Figs 5 to 9 due to the added Fig.1) have been changed to include the results of the sensitivity study with respect to the explained variance relation between reconstruction and reference curve.

**Review 1**

We thank the reviewer for the very helpful insights and remarks. Those are in italics

*Netzel et al. present a new method for pollen-based quantitative climate reconstructions based on the 'mutual climate range' approach. They develop an algorithm using Bayesian statistics to overcome the disadvantages of other methods such as age model uncertainties, criteria for plant taxa inclusion/exclusion, and human-induced impact on natural vegetation. The method is applied on a previously published palynological dataset from Lake Kinneret (Eastern Mediterranean region) that spans the past c. 9 kyrs suggesting a mean winter temperature of c. 10 oC throughout this time interval and a mean annual precipitation of about 400–700 mm.*

*Although I am not an expert on the development of pollen-based climate reconstruction techniques, I have read this manuscript with great interest because of the necessity for robust and precise reconstructions of past climates. This is a well-structured manuscript and generally clearly written. In my view, however, there are several shortcomings that preclude publication in 'Climate of the Past' in its current form.*

Necessary citations from the original text are highlighted in color

And followed by the corrections/remarks/additions to the text.

**Main concerns:**

*1. The authors argue that the comparison of the reconstructed climate parameters with those provided by other reconstruction methods will test the reliability of the results (Lines 67-68). I struggle to understand the logic behind this approach. For example, if the here-reconstructed values are comparable with those provided by other methods, the question that arises is why a new method is needed. If the results are not comparable, then the question that arises is how one can test which reconstructed values are more realistic. In my view, independent (i.e., non-pollen-based) climate reconstructions should be used to check the reliability of the here-reconstructed climate parameters. This aspect needs to be elaborated on in a revised manuscript.*

Line 67- 68: We will use these as a comparison for our quantitative statements to show similarities and differences and to check whether our new approach fits into the existing knowledge and thus provides realistic results.

We changed and added:

A comparison is possible following the results from previous reconstruction studies of paleoclimate information in the Dead Sea region (Litt et. al, 2012). These exhibit certain deficits like biases at recent time slices or extensive variability during Holocene times. Details will be discussed below. The aim of the present study is to evaluate the potential of those mentioned, additional environmental data to enhance similarities and reduce differences between reconstructions at LK based on the previous methods and the information content of the additional qualitative data. The general approach is a Bayesian statistics based data assimilation of the new data (likelihood) into the previous reconstruction forming the prior. The resulting posterior will not only provide a most probable reconstruction of the paleo climate state given both type of input data but also an uncertainty estimate. The latter allows a comparison of the prior reconstruction with the posterior one and an assessment of the gain of information by the assimilation without the need of independent data (e.g. from non-pollen data like stable isotopes). The theoretical concept presented in this study readily extents to the inclusion of those independent data, which is a task for future work. In addition, already available data on lake level fluctuations can be used as independent proxies at least for precipitation changes for comparison with pollen-based reconstructions (Lake Kinneret: Hazan et al. 2005; Vossel et al. 2018; for regional scale Dead Sea: Stein et al. 2010, Litt et al. 2012).
* * *
*2. The authors consider 'a priori that the climate reconstructions should explain about 50 % of the variance of the respective reference curves' (Lines 182-183). If I understand correctly, this threshold value of 50 % is regarded as sufficient to address the human influence on vegetation dynamics. In my view, this is an important point that needs to be explained further. How is this 'about 50 %' defined? Is it based on statistical analysis or empirical observations? And how would the reconstructions be affected if this threshold value was higher or lower? Clearly, the human influence on natural vegetation has gradually intensified during the Holocene, and the manuscript does not explain how the new method elaborates on this. As such, the statement that the method 'helps to reduce the human impact on vegetation during the reconstruction process' (Line 412) is not fully substantiated.*

==Line 182/183: To account for such uncertainties, we specify a priori that the climate reconstructions should explain about 50% of the variance of the respective reference curves.==

In particular, the influence of humans on the vegetation around the lakes studied during the mid- to late Holocene complicates the interpretation of these curves. To account for similarities and to reduce differences the above mentioned Bayesian statistics based assimilation of the AP data is based on the notion that the regression of the climate reconstruction to the reference AP curve should explain an anticipated amount of variance. Inline with the general Bayesian approach that amount is not a fixed number but described by a most probable value and an uncertainty. By performing a sensitivity analysis the most probable value was varied in the steps 25% , 50% und 75% and a typical uncertainty of 20% obeying the constraint that the explained variance can only vary between 0 and 100%. This gives the model the ability to capture a sufficiently large range of R2 (Netzel, 2023a). For the 50% case such a proposal can be described by a beta distribution with shape parameters (3,3) (for 25% it is (,), 75% (,))

Here formula (5)

This approach allows the climate reconstruction to the follow the trends of the AP references (or any other non-pollen data set) under the assumption that the biome information B provides sufficient variability through the probabilistic transfer function. On the other hand, additional moderators like human influence are allowed. The aspects of human impact related to the climate reconstruction method (line 412) will be explained in more detail the discussion chapter (see also response to 4).
* * *
*3. The mean winter temperature reconstructed values (Figure 9a) show almost no variability over the past 9 kyrs. Instead, a quasi-stable temperature of about 10 oC is suggested to have prevailed for such a long period of time. Arguably, the method is not sensitive enough to capture changes in temperature, despite the fact that the temperature reconstructions are generally easier than e.g. precipitation reconstructions. I suggest the authors to compare their temperature reconstructed values with other temperature records from the region (also non-pollen-based) and discuss vigorously what the limitations of the method are, and why the here-presented results provide meaningful and reliable climate reconstructions.*

Note that the presented method reconstructs a full probability density pdf of the joint Dec-Jan-Feb mean temperature and the annual precipitation sum at a given age. The apparent smoothness of the Dec-Jan-Feb mean temperature in Fig. 9a (commonly referred to as winter temperature on the Northern Hemisphere) results if one concentrates on the median of the reconstructed pdf without considering the inherent variability. The median temperature is that temperature value which divides the reconstructed temperature range into two equal probable intervals from which individual realisation of the NH-winter temperatures have to be drawn at random. This randomization introduces additional variability in the time series but requires the specification of the autocorrelation in time which is not modelled yet in the present method. The effects of such randomization in the climate field reconstruction of Holocene temperature in Europe has been demonstrated by Simonis et al (2012) (citation added) . The comparison of the present reconstruction with other temperature reconstruction eg based on non-pollen data can only be done if these two type of information (the most probable or median value plus the implied variability) are quantitatively  available. The comparison is possible for the Dead Sea reconstruction from Litt et al. 2012. The results will be discussed below.
* * *
*4. The mean annual precipitation reconstructions (Figure 9b) mirror the variability in the arboreal pollen %, which in turn they predominantly reflect changes in the Olea percentage (compare Appendix B). As such, the precipitation reconstructions are misleading because Olea  is closely related to agricultural practices in the Eastern Mediterranean region. On this basis, there is a very strong human component in the reconstructed values that appears to obliterate the natural climate variability. This view is also supported by a close look at the timing of the Cichorioideae % peaks (compare Appendix B), which are also considered indicators of human-induced land use changes. Specifically, the precipitation drops at 4 and 3.2 cal. kyrs BP, which the authors attribute to short-term climate changes related to the Bond events (Lines 371-372), conspicuously coincide with Cichorioideae % peaks. As such, the precipitation reconstructed values may also represent a human-*

*induced signal rather than climate variability. As for the temperature, the precipitation reconstructions provided by this new method should be compared with other precipitation records (also non-pollen-based) and vigorously discussed in a revised manuscript.*

To be added in the discussion chapter near Lines 371-372

On the other hand, the decrease in oak pollen of 4 and 3.2 cal ka BP could be related to the Bond events of 4.2 and 3.2 associated with droughts in the Levant.

We added: Olea europeaea is an integral part of the Mediterranean vegetation zone, even as an indicator species for the current geobotanical distribution of this biome (see Langgut et al. 2019).Olea also grows as a cultivated tree mainly under Mediterranean climate conditions. When olive groves were planted in the past, the Mediterranean oak forests, which were predominantly deciduous, had to be cleared (e.g. Q. ithaburensis). Oak trees were therefore replaced by olives and vice versa (see Fig. B1 and B2). Both species have a similar chance of being recorded in the pollen record (high pollen producer based on wind pollination). It is also noteworthy that the bivariate conditional probability density functions (likelihood functions) of winter temperature (DJF) and annual precipitation are very similar for both species (see Neumann et al. 2007).

The subfamilies Cichorioideae and Asteroideae, both belonging to the Asteraceae family), are components of the Irano-Turanian steppe vegetation. They might also occur in the anthropogenic influenced Mediterranean vegetation zone (batha, garrigue). However, it must be stressed that the Cichoioideae peaks appear in a phase which was less influenced by human impact (Miffle Bronze Age after the decrease of Olea cultivation and increase of Q. ithaburensis type). Therefore we assume a stronger climate signal related to Cicorioideae peaks (dryer conditions).

And after line 444 we added: It is interesting to note that the reconstructed Dead Sea lake level curve as an independent proxy for precipitation (Stein et al., 2010) correlates very well with the pollen-based paleoclimate reconstruction (Litt et al., 2012). However, it must be stressed that the older reconstruction method based on a Bayesian Biome Model has some weaknesses compared to the new approach which are not detectable by the correlation, namely systematic shifts (biases) with respect to present climate, e.g. the mean Dec-Jan-Feb temperature in Litt et al. (2012) is clearly to low due to the inclusion of temperature values of the Mediterranean vegetation zone in the northern part of the study area.

**Other comments:**

*Line 7: Unclear phrasing '…that map climate variable to biome distributions'.*

Furthermore, we introduce a systematic way to establish transfer functions that map climate variables to biome distributions.

We introduce a systematic machine learning based way to establish probabilistic transfer functions which connect spatial distributions of temperature and precipitation to the spatial presence of specific biomes. Mean Dec-Jan-Feb temperature grid point values and the annual sum of precipitation at that grid point are used to classify the presence or absence of the biomes.

*Line 13: Do you refer to arboreal pollen percentages? Please specify here and throughout the text.*

Here, a priori information on the recent climate in this region and data on arboreal pollen from this lake are used as boundary conditions.

Here, a priori information on the recent climate in this region and data on arboreal pollen percentages from this lake are used as boundary conditions.

… and changed throughout the text

*Line 23: Add references.*

In the last few decades, a lot of reconstructions were published, which showed the advantages and disadvantages of the respective methodologies.

We added: In the last decades, several reconstructions were published (e.g. Neumann et.al, 2007; Langgut et. al. 2013; Litt et.al (2012), Langgut et. al. 2019), which showed the advantages and disadvantages of the respective methodologies.

*Lines 44-45: It is unclear what the previously application of the BBM approach on the Lake Kinneret dataset has shown. Please expand the text and explain what is the relevance for this study.*

Thoma (2017) applied the BBM to Lake Kinneret (LK), also known as the Sea of Galilee, with the result that the two biomes used showed too little variability and suggested an expansion to at least three biomes.

We added and replaced into: A first application to Lake Kinneret (LK), also known as the Sea of Galilee, is shown in the work by Thoma (2017). He used the time series information of the two major biomes which can be deduced from the LK core. The resulting BBM based paleo climate reconstruction did show too little variability in temperature and precipitation suggesting that at least a three biome model as basis for the BBM should be used. Together with changes in derivation of the transfer functions, the assimilation of the present time and arboreal pollen percentage time series a three biome approach (Mediterranean, Irano-Turanian, Saharo-Arabian) plus a virtual biome necessary for the actual biome vs. climate transfer function calculations will be implemented in the following. The virtual or undefined biome summarizes all neighbouring biomes in the Levante not covered by the three physical ones. It is especially needed in in the neuronal network NNET method but to compare these results it is also used in the three other methods. Finally, the BBM also allows reconstructions based on prior climate data. These come, for example…

*Lines 74-76: What is the relevance of the information on the lake's characteristics for this study. Please explain or delete.*

The location of Lake Kinneret is marked with a black dot in Fig. 1 (a). LK is a warm, monomictic and meso-eutrophic inland lake being part of the Jordan river catchment and its lake level varies between 209 and 215m below mean sea level. It has a maximum water depth of ca. 42m and a surface area of ca. 169km2 (21×12km at the maximum). The watershed area comprises 2730km2 (Berman et al., 2014).

We deleted some information on lake´s characteristics and replaced into:

Lake Kinneret in Galilee (Israel) is part of the Jordan river catchment. It has a maximum water depth of ca. 42 m and a size of 21 x 12 km (Bermann et al., 2014).

To continue:  (New) Fig. 2 (a) and (b) show …

*Lines 86 and 92-95: The Kinneret basin cannot be seen in Figure 1, and by extension, the prevalent climate conditions and vegetation biomes in the study area. As such, Figure 1 has to be redrawn.*

We added a new Fig. 1 (catchment area southern Levent including Lake Kinneret and Dead Sea Basin, biome distribution, annual precipitation).

Older Fig. 1 is now Fig. 2:

This remark by the reviewer is difficult to understand: the maps have a (CRU) grid size of 0.5° * 0.5° (~50 * 50 km) with the maximum extension of LK being 12 * 21 km, thus well represented by the black dot in the three maps in Fig. 2 (Fig. 1 in the old version).

*Line 94: The vegetation and climate characteristics of the Saharo-Arabian biome are not presented, despite this biome is discussed in both the 'results' and 'discussion' sections. Please also explain what is the 'unspecified biome' and why it is worth mentioning in the text as long as it not found in the catchment area of Lake Kinneret.*

We added: Saharo-Arabian desert vegetation occurs in the southern part, where the mean annual precipitation falls below 100 mm. It is a vegetation type with sparse plant cover and low diversity. Important representatives of the Saharo-Arabian vegetation are Zygophyllum dumosum, Retama retam, Tamarix nilotica, Atriplex halimus and other Amaranthaceae. Sudanian vegetation occupies tropical oases of the Jordan Valley. Mainly trees and shrubs such as Maerua crassifolia, Acacia radiana/Acacia tortilis, Balanites aegyptiaca, and Ziziphus spina-christi compose this vegetation type (Zohary, 1962).

*Lines 155-162: I don't understand how the use of a 50 years grid (which is defined based on the 51 years average temporal resolution of the pollen record) addresses the 'full age uncertainties' (see line 50). How does the new method elaborate on the changing sedimentation rates in the lake over the past 9 kyrs?*

In this case, the target variable is AP and is shown in panel (a) as a function of sediment depth. The mean age difference between the studied core intervals of 11 cm is 51 years. Thus, we define a regular temporal grid of 50 years via P(D|A), resulting in a total of 181 age steps. Applying these to the data from panel (a) using Eq. 2, we get the result of the new age–depth transformation depicted in (b). In contrast, the orange line shows the result when the plant data in terms of depth are linked to the mean age data from the age–depth model. So far, this is a very common procedure (e.g. Litt et al., 2012; Schiebel and Litt, 2018; Torfstein et al., 2015; Neumann et al., 2007;  Miebach et al., 2019; Seppä et al., 2005). The strongly fluctuating behaviour of this AP curve indicates an overfitting result,

which makes interpretations difficult. With this new technique, we can circumvent such problems and have eliminated the first disadvantage mentioned in the introduction.

We changed into:

In this case, the target variable is AP-percentage and is shown in panel (a) as a function of measured sediment depth. Upon using the depth-age relation of the most probable age at a given depth the

mean age difference between the studied core intervals of 11 cm thickness is 51 years. Thus, in a first step we define a regular temporal grid of 50 years resulting in a total of 181 age steps between 0.0 and 9 kyr BP. Based on the results of the full probabilistic analysis of the sedimentation-time relationship available from the rbacon package the newly added function *Bacon.d.Age* determines those depth samples which belong to a given age with a probability between zero and one including the changing sedimentation rates in the lake over the past 9 kyrs modelled internally in rbacon. Applying equation (2) then weights depth either with near zero or with a finite probability value given an age on the 50 year time grid between 0 and 9 kyr BP. By the integration in Eq. (2) the depth related probabilistic reconstructions are data-dependent stretched or compressed in time and smoothed in time over that interval with finite, non-zero probabilities of depths given the desired age. By this procedure the approach addresses the full age-depth uncertainty. Since age is a given variable (by this not anymore a random variable as it is in the conditional probability of age given the sediment depth) in principle any time stepping (10, 25, 100 yrs) could have been chosen but the 50 yr time step is to some instance determined by the data set itself.

Applying it to the data from panel (a) using Eq. 2, we get the result of the new age–depth transformation depicted in (b). In contrast, the orange line shows the result when the plant data in terms of depth are linked to the mean age data from the age–depth model. So far, this is a very common procedure (e.g. Litt et al., 2012; Schiebel and Litt, 2018; Torfstein et al., 2015; Neumann et al., 2007; Miebach et al., 2019; Seppä et al., 2005). The strongly fluctuating behaviour of this AP curve indicates an overfitting result, which makes interpretations difficult. With this new technique, we can circumvent such problems and have eliminated the first disadvantage mentioned in the introduction.

*Line 194: Please explain what do you mean with 'specific expert knowledge' and how this can be used in a quantified manner that is required for the climate reconstructions.*

$P(Cp \mid \cdot)$ gives us the ability to constrain the reconstructions based on additional climate information from the past. These can be, for example, other local reconstructions, paleoclimate simulations, or specific expert knowledge based on vegetation studies.

Changed into: $P(Cp \mid \cdot)$ gives us the ability to constrain the reconstructions based on additional climate information from the past. These can be, for example, other local reconstructions, paleoclimate simulations, or even subjective expert knowledge based on vegetation studies. Often, the latter is a common approach in classical Bayesian statistical analysis (see new citation Berger, 2013). In the simplest case it would be a subjective probabilistic statement with a number between zero and one (but excluding those) about the climate state Cp given the age and the proxy data.

*Lines 210-213: I am not sure if I understand this correctly. Do the authors mean that the method cannot be applied for long time periods that would require changes in the taxa weights? Please explain in more detail in order to make clear to non-experts any limitations of the method (e.g., continuous reconstructions for a whole glacial-interglacial cycle).*

First, we assume that the parameters ω and ψ are a priori independent of each other. Then we state that P is independent of ψ if no C is given. Finally, the updated taxa weights P(P | ω) are determined under the assumption that they are conditionally independent of A and thus hold for the entire reconstruction period. At this point, taxa weights could be split temporally based on additional prior

information, so that they differ for specific time periods (e.g. glacials/interglacials). This approach is not explored further in this study and could be included in future work.

Changed into: First, we assume that the parameters ω and ψ are a priori independent of each other. Then we state that P is independent of ψ if no C is given. Finally, the updated taxa weights P(P | ω) are determined in the present case from the AP-percentages under the assumption that they are conditionally independent of the age A. This means the additional data used to update the weights are assimilated over the entire reconstruction period. At this point, it is possible to introduce additional prior information for time continuous reconstructions across a full glacial / interglacial cycle. The taxa weights updating could be split according to that temporal information  so that after assimilation they differ for specific time periods.  This approach is not explored further in this study and could be included in future work.

*Line 328: Explain for non-experts what is 'C++' and 'standard CPU'.*

Due to the large number of parameters, we decide to generate 1 million MCMC samples. To make such a reconstruction as fast as possible, C++ is used. Thus, a reconstruction on a standard CPU takes only about 40 seconds.

Changed into: Due to the large number of parameters, we decide to generate 1 million MCMC samples. This makes the numerical problem difficult to solve fully in a R (or python) programming interface. Therefore as much as possible subroutines are implemented in the compiler language C++ and embedded into the R code.  By this approach the reconstruction model can be implemented on a standard laptop or stand-alone PC with commercially available, standard central processors and uses about 40 – 60 seconds for the MCMC samples and their evaluation.

*Line 364: Higher than what?*

Schiebel and Litt (2018) assume climate change towards higher precipitation, which is also confirmed by our reconstruction.

We changed and added: Schiebel and Litt (2018) assume climate change towards higher precipitation compared to the previous time slice (9-7 cal ka). In addition, Hazan et al. (2005) and Vossel et al. (2018) describe a high Kinneret lake level during the Chalcolithic and Early Bronze Age, which is also confirmed by our reconstruction.

**Review 2**

*Citations from  RC2 are in italics*

*Netzel et al consider a series of developments to the Bayesian Biome Model (BBM) of Schoetzel et al (2006), a Bayesian hierarchical paleoclimate reconstruction approach which incorporates a probabilistic interpretation of mutual climatic range which is applied by assigning taxa to biomes and considering the relative influence of two or three biomes in the*

*fossil assemblage. This model has previously been applied to a series of reconstructions, most recently including Lake Kinneret (Thoma, 2017). The authors identify a series of weaknesses in the existing approach (which may apply more generally than to only BBM). These can be summarised as neglect of age uncertainty, neglect of effects of human impacts, assumption of fixed spatiotemporal species-climate relationships and the need for user-defined (subjective) decisions in respect of taxa selection, parameter values and choice of model choice. They address these through a series of modifications to the published method.*

*My initial reaction was that the approach was rather scattergun, addressing a range of different and unrelated reconstruction issues, and that it might be of limited scientific value because it is unclear what it teaches us is needed to progress. Overarchingly, the paper proposes and implements a series of modifications together with a single reconstruction of Lake Kinneret from this revised model which is compared against a published reconstruction on a core from the Dead Sea (Litt et al 2012). For me, demonstrating that these two reconstructions are consistent is not very convincing because it does little besides suggest that the modifications haven't made things worse, and in fact have made little material difference?*

*In part this work is an attempt to automate some ad hoc decisions in the interests of reproducibility and ease of use (and quality of reconstructions). But even for this motivation I think a more complete validation is needed to justify choices and additional complexity.*

*In my opinion the paper should be restructured and expanded to address the question of which, if any, of these modifications are materially useful and how they influence the reconstruction. Firstly, I think the paper should be more clearly set out to identify which modification relates to which weakness e.g. with clearly headed subsections in both the methods and results that relate back to the weaknesses identified in the introduction. More importantly, each modification should be analysed and discussed in isolation, for instance by starting with the baseline model (of Thoma 2017?) and performing a reconstruction with and without that modification. Something like this is needed to isolate and understand the effects of each modification, not only on the reconstructed value but also on the uncertainty associated with the reconstruction. Note that I am deliberately using 'modification' and resisting 'improvement' because I am not confident this has been demonstrated yet.*

We thank the reviewer for these very helpful  insights and remarks.

The paper can indeed be ordered according to the subtopics coming from the evaluation of past reconstructions using pollen based BBM, the central point of the paper is that this can all be done under the umbrella of Bayesian statistics namely starting with

(1) Quantitative inclusion  of age uncertainty,

and then proceeding to

(2) Evaluate effects of potential human impacts upon the climate reconstruction
(3) More flexible treatment of the spatial taxa-climate relationships (transfer functions)
(4) Include on the prior level user-defined (subjective) decisions in respect of taxa selection, parameter values and choice of model choice.

These new guidelines are discussed first and then implemented based on the experiences from BBM eg in Thoma (2017) or Litt et al 2012.

Regarding the point that "demonstrating that these two reconstructions are consistent is not very convincing because it does little besides suggest that the modifications haven't made things worse" is a look at the final outcome only. Rather the whole chain of implementations of points (1) to (4) under the Bayesian thinking provides a clear advantage over past attempts. The central one is indeed the point made by the reviewer "to automate some ad hoc decisions in the interests of reproducibility and ease of use and raise the quality of reconstructions". These points will be addressed in the motivation and at the start of the discussion chapter (..and – as a side remark -- the numerical solution by Markov Chain Monte Carlo underlines the view of a chain of implementations).

We included explicitly: indeed the point made by the reviewer "to automate some ad hoc decisions in the interests of reproducibility and ease of use and raise the quality of reconstructions".

One point which makes the comparison with the previous reconstruction difficult is the introduction of the probabilistic age-depth relationship and its influence upon the reconstruction. As already outlined in the manuscript previous reconstructions use a specific version of the conditional probability of age given the sediment depth. Very often this is the maximum (mode value) of that conditional probability or the estimated expectation of age under the conditional probability. The version we put forward utilizes a very different conditional probability namely that of sediment depth with a give age, for details we refer to our remarks with respect to the "Lines 155-162" comment by reviewer 1 (see above). The use of Eq, (2) in the proposed chain of changes leads to two very different data set which cannot directly be compared on neither the sediments depth axis nor the time axis. This also holds for other data sets based on sediment cores, it would not hold for data from climate model simulation which could give hints of the quality of the reconstructions. But this is well outside the present aim of the manuscript.

This point is explicitly discussed in the respective chapter on age modelling

**Specific points**

*Section 3.3.2 discusses the age model and compares pollen percentages due to the revised age model, which has the effect of smoothing the signal. Can this plot instead / in addition plot a comparison of reconstructed climate? I suppose the signal must be smoother, but how much, and what are the effects if any on the uncertainty? Smoothing is only useful to the extent the original variability is spurious, can you justify this - why does "strongly fluctuating behaviour of this AP curve indicates an overfitting result"? Bronk Ramsey developed a Bayesian carbon dating approach OxCal which incorporates the constraint that increasing depth implies increasing age, and which provides useful information through the calibration curve because atmospheric C14 varied over time. Could you comment on this, perhaps only in your response if that's sufficient, my knowledge of this is rather old and perhaps outdated! I would be interested what effect using the Bronk Ramsey approach might have on your age depth profile.*

Starting with the eldest problem: the Bronk Ramsey Bayesian OxCal results are compared in the original bacon paper Blaauw,M. and Christen, J. A. (2011). As such the bacon model for age-depth analysis is a more elaborated and advanced method than those acc Bronk Ramsey.

The usual application of either model is the analysis of the derived (posterior) conditional probability of age given a sediment depth. However, here in Sect 3.3.2 we discuss the conditional probability of sediment depth given a specific age, or in other words if an age (on a time grid of 50 years between 0 and 9000 yr BP) is prescribed, which sediment depth belong with a finite probability (clearly larger than 0) to that age. If these probabilities are know Eq.(2) tells the reader that the posterior data set is evaluated by that integral, there is a data dependent stretching or compressing of the sediment depth samples together with a data dependent smoothing of the respective reconstruction values obtained from each sediment depth. The choice of arboreal pollen percentage in Fig. 2 is at this point for illustrative purposes, any other sediment depth defined data set can be used instead, in the discussion part we will use the reconstructed probability density function of the climate variables temperature and precipitation. But the course of the analysis of this way to include age-depth uncertainties is identical. The new conditional probability of sediment depth given age is part of the new bacon version (rbacon) described in Blaauw, M., Christen, J. A., and Aquino L., M. A. (2020).

*Reg. I suppose the signal must be smoother, but how much, and what are the effects if any on the uncertainty?*

This is indeed a relevant question, which is not yet solved in its completeness. It needs the repeated simulation of random white noise at the sediment depths, the application of Eq. 2, computing the Fourier spectrum of the resulting stretched/compressed/smoothed output unstructured noise (which can only done on the regular time grid)  and the squared averaging of the resulting spectrum to derive the equivalent of the gain function known from classical Fourier transform.  This general outline shows that several data dependent steps are involved ranging from the rbacon internal modelling, to the input of the sediment depth and the C14 anchor data. This makes the result strongly dependent on the specific original data sets and actually to a result of the extended rbacon modelling with a potential addendum in rbacon to be discussed with the bacon authors.

*Smoothing is only useful to the extent the original variability is spurious, can you justify this - why does "strongly fluctuating behaviour of this AP curve indicates an overfitting result "*

The standard use of the age depth relationship is already incorporated in Eq.2 (therefore the Bayesian statistics approach is more general as the standard way of age depth calculation) and illustrated by the orange line in Fig.2. It is achieved for a given age by selecting a **single** sediment depth with probability 1  e.g. , that depth at which the conditional probability of depths given the age is at a maximum (mode value) and then computing formally the integral. No information about the age depth related uncertainty is used, only one sediment depth is determined for a given age, clearly a case to be identified as "overfitting".

This point is now explicitly mentioned in the text.

*You "specify a priori that the climate reconstructions should explain about 50 % of the variance of the respective reference curves". This seems a rather ad hoc assumption, could you e.g. explore the sensitivity of the reconstruction to this assumption, perhaps with two extreme (but justifiable) choices?*

Apparently this approach is not clearly described in the original manuscript as both reviewers refer to it. At this point we would refer to the answer given above to reviewer 1, comment to Line 182/183.

*What effect does the prior have on your reconstruction? Again, a comparison with and without the CRU prior seems appropriate. This is another modification that will presumably smooth your reconstruction, is this smoothing justified? I don't really understand why, given that climate change/variability is usually the thing of interest, you would want to inhibit that by applying a prior that assumes no change?*

As outlined in the answer to RC1 above (the line 182/183 one) we include the sensitivity study of the prior choice analysing the reconstructions for the prior choices 25%, 50% and 75%. The results are discussed in the respective chapter and clearly show the anthropogenic contributions (from Olea vs Quercus) onto the reconstruction. This is mainly found in the new Fig 6, Fig 7, Fig 8, Fig 9 and Fig 10. We also discuss the point how the prior distribution of the $R^2$ (which does not only involve the mode values 25,50 and 75%, but also an assumed uncertainty of 20% under the Beta distribution, a table is added with the necessary Beta shape parameters depending on the means and variance) changes into the posterior distribution under the influence of the original BBM reconstruction and the additional taxa based reconstruction to match the arboreal pollen percentage line with the posterior realisation of $R^2$. From those three posterior distribution (see updated Fig. 8) one can conclude that the choice of the prior beta distribution $R^2$ with a mode value of 50% and 20% uncertainty leaves enough degrees of freedom for optimizing (data assimilating) the arboreal pollen percentage time series by varying the influence of single taxa because the posterior density shifts its model value to larger $R^2$ plus enough uncertainty to avoid the collapse to only a very few selected taxa (which happens at both 25% and 75%). Further, the two runs with 25% and 75% prior mode values exhibit only small changes of the posterior mode values, in case of the 75% even a slight reduction.

The additional CRU based prior information is applied to the whole reconstruction and serves as a bias correction together with the new transfer functions e.g. shifting the Litt et al 2012 DJF temperature reconstruction for recent time slices from unrealistic 0°C to more realistic 10°C without affecting the temporal variability of the complete time series.

All points are summarized and discussed in the text now.

*A machine learning competition is used which selects the NNET algorithm as that maximises the 'balanced accuracy' under cross validation. I would like to see a comparison of the reconstructions from the four approaches. Are they quantitatively distinguishable, i.e. is the additional complexity of SMOTE justified? Are they qualitatively distinguishable, for instance because they behave differently under extrapolation beyond the training set, so that BA is an insufficient metric to decide the "best" model?*

This is already (but apparently not in full completeness) summarized in Fig. 4. Fig.4a shows the gain of all four methods which they achieve with respect to the SMOTE approach. Balanced accuracy is a performance metric for two by two contingency tables estimating the joint probabilities of real (test) data occurrence vs that derived from either of the four classification algorithms taking into account that one combination (here true negative case absence of a biome vs predicted absence of that biome at all grid points in the study area, Fig.1 and 2) is much larger than the three other possible cases ( true positive case presence of a biome vs predicted presence of that biome, false positive true presence of a biome vs predicted absence of that biome, false negative absence of a biome vs predicted presence of that biome), a so called unbalanced data set which is still used for the test data. SMOTE is a

way to re-balance the input data in the training phase such that actual number of observed biome grid points is artificially enhanced to match the overall number of available grid points. The effects of SMOTE is very well documented by the increase of the balanced accuracy from roughly 0.5 (non-SMOTE) to about 0.92 for the SMOTE treated training data sets. In terms of BA all four classification methods including the classical QDM behave similar, the choice of NNET is justified in the text. The comparison of the four different ways to derive the transfer functions in the final reconstruction is in principle possible but requires a 4 by 4 comparison of reconstructed probability densities to measure the full information content of the pdf's e.g. by using an entropy measure and an evaluation of that in terms of the common signal of the reconstructions, the added value of either reconstruction over the other etc to allow for a clear and reproducible discussion. A three by three approach for Gaussian pdf's (which is not applicable in the present case) is described in Glowienka-Hense et. al (2020): Glowienka-Hense, R., Hense, A., Brune, S., & Baehr, J. (2020). Comparing forecast systems with multiple correlation decomposition based on partial correlation. Advances in Statistical Climatology, Meteorology and Oceanography, 6(2), 103-113.). Such a comparison being indeed useful and necessary is currently beyond the scope of this manuscript. A classical comparison e.g. based on the mean or median time series would ignore specific aspects of the full reconstruction model.

*Are they qualitatively distinguishable, for instance because they behave differently under extrapolation beyond the training set, so that BA is an insufficient metric to decide the "best" model?*

No, the experiments (details can be found in Netzel 2023a) indicate no significant differences when applied to the left-out part of the data set (test part). We mention this at the appropriate place in the text.

*I wasn't clear, is it intended that the ML competition is run on any new data set, or are you concluding NNET is the best model in general for this problem? i.e. is does your algorithm incorporate the competition or does it apply NNET by default?*

Yes, one conclusion of the current manuscript is that the ML competition needs to be run on any new sediment core after, actually the conclusion is that the full model with its 4 steps needs to be used to arrive at pointwise data sets which can serve as input to climate field reconstructions at a given age (time slice) and as sequences of several time slices e.g. the full Holocene or the transition from the last glacial maximum into the Holocene etc. This point is discussed in the conclusions section.